**Subject Category:**
Biology (whole organism)

biomechanics/ecology

predation, coral reefs, teeth, morphology, ecomorphology, functional morphology

**Author for correspondence:**
Michalis Mihalitsis
e-mail: mike.mihalitsis@my.jcu.edu.au

# Functional implications of dentition-based morphotypes in piscivorous fishes

## Michalis Mihalitsis[1,2] and David Bellwood[1,2]

[1]College of Science and Engineering, and [2]Australian Research Council, Centre of Excellence for Coral Reef Studies, James Cook University, Townsville, Queensland 4811, Australia

MM, 0000-0001-7270-6879; DB, 0000-0001-8911-1804

Teeth are crucial in elucidating the life history of vertebrates. However, most studies of teeth have focused on mammals. In heterodont mammals, tooth function is based on tooth shape and position along the jaw. However, the vast majority of vertebrates are homodont, and tooth function might not be based on the same principles (in homodonts, tooth shape is broadly similar along the jaw). We provide a quantitative framework and establish dentition-based morphotypes for piscivorous fishes. We then assess how these morphotypes relate to key functional feeding traits. We identified three broad morphotypes: edentulate, villiform and macrodont, with edentulate and villiform species displaying considerable functional overlap; macrodont species are more distinct. When analysing macrodonts exclusively, we found a major axis of variation between 'front-fanged' and 'back-fanged' species. The functional interpretations of this axis suggest that tooth-based functional decoupling could exist, even in homodont vertebrates, where teeth have similar shapes. This diversity is based not only on tooth shape but also solely on the position along the jaw.

## 1. Introduction

Vertebrate teeth have been studied for centuries. Their importance in elucidating the life history of organisms has been demonstrated in multiple fields, from palaeontology and evolution to ecology. Usually, the focus is on biomechanics, morphology and/or behaviour (e.g. [1–6]). However, most studies of vertebrate teeth have been focused on mammals [7–11]. Lower vertebrates, although more speciose, have received less attention.

Fishes and, more specifically, teleosts constitute over half of all vertebrate species [12]; however, our understanding of their oral tooth morphology was for a long time primarily at a descriptive

level: small/large, conical/villiform/molariform [13]. In the last decade, however, research has begun to elucidate the morphology and potential function of several aspects of fish dentition [6,14–20]. These studies have provided invaluable information on how the tooth morphology of fishes may influence their feeding capabilities. However, if we are to link tooth functional morphology to ecological functions and, more specifically, to how fishes feed in their environment, there is a need to identify functional groups based on full dentition morphologies rather than individual teeth.

The limited number of more quantitative descriptions of fish dentition, when compared with mammalian dentition, is not without a good reason. First, mammals only replace their teeth once throughout their lifetime (diphyodont). Fishes, by contrast, along with most other lower vertebrate lineages, constantly replace their teeth (polyphyodont) [10,21]. Second, fishes display significantly higher variance in the distribution of their oral teeth along their jaw when compared with other vertebrate groups [10]. For example, mammals primarily have canines on the anterior part of their jaws, and no mammalian species has more than one canine in each quadrant (upper left versus lower right, etc.) [10]. It is, therefore, possible to classify mammalian dentition based on the number of teeth of each type using dental formulas. For example, humans have the dental formula I(2/2),C(1/1),P(2/2),M(3/3), where letters indicate tooth type (I, incisors; C, canines; P, premolars; M, molars) and fractions indicate the number of teeth on upper and lower quadrants. For fishes, this would be a herculean task, given the extent of variation in form and number. Furthermore, compared to a mammal, like humans, which as seen above have 32 teeth, fish can have thousands of teeth in their mouths [10]. Finally, tooth function in mammals is based on tooth shape and location along the jaw (e.g. canines = large conical teeth anteriorly in the jaw versus molars = relatively flat teeth located posteriorly). Unlike mammals that have different shaped teeth (heterodont) [10], fishes and other lower vertebrates typically have similarly shaped teeth (homodont) [22]. However, these descriptive terms, homodont and heterodont, need to be interpreted with caution, as the term 'different shaped teeth' can sometimes be misleading. In this study, we follow Liem *et al.* [23], who noted that '…in the majority of vertebrate species, the teeth, although they may differ in size, are structurally alike, a condition called homodont'. This issue was discussed by D'Amore *et al.* [24], who noted the need for a broader evaluation of tooth form and function.

These terms (homodont/heterodont) offer definitions that provide a coarse framework for the comparative analysis of tooth form. However, tooth form may not be the only trait determining tooth function [14,15] within lower vertebrates. Overlap among groups is inevitable. There is clearly a need to expand our frame of reference from individual tooth form and function to the entire dentition morphotype and its functional implications.

The importance of establishing such a framework, however, is that it will enable us to begin to link morphological traits with functional morphology, i.e. establishing a functional link between certain anatomical features and how they help the organism perform a specific task (e.g. feeding) [25–27]. These characters/traits, can, in turn, be linked to the way organisms interact with, and more importantly influence their surrounding environment [28,29].

One group of organisms displaying high morphological diversity, and thus making them an ideal study group, are piscivorous coral reef fishes [30]. This group of fishes displays high morphological diversity related to feeding traits such as gape size [30,31]. It has been suggested that this diversity may reflect the potential for niche partitioning on the prey of different sizes, or different feeding modes [30]. However, before beginning to ask such questions, there is a need to first delineate the various dentition morphotypes found within this functional group.

In our study, we provide a quantitative comparative framework of piscivorous fish dentition types and use the data to identify distinct morphotypes. We then show that these morphotypes are linked to key functional feeding traits. Finally, we show that the location of the largest teeth along the jaw can have biomechanical and, therefore, functional implications. We argue that some teleost lineages might have evolved a type of functional decoupling where similarly shaped teeth could have a different function, based solely on their position along the lower jaw.

# 2. Material and methods

## 2.1. Morphological measurements

In total, we measured 61 freshly thawed specimens of 29 piscivorous teleost fish species (mean = 2.1 individuals per species). Standard length measurements (SL) were taken using calipers or measuring tape for larger specimens. Ontogenetic shifts in dentition were minimized by measuring only subadult

and adult specimens. Vertical and horizontal oral gape distances were measured using scissors, following the methodology and definitions of Mihalitsis & Bellwood [32]. Specimens were then displayed perpendicular to a camera and photographed, first mouth closed, then mouth open (maximal jaw depression). Upper jaw protrusion was measured as the difference between the distance between the tip of the jaw to the anteriormost point of the eye with the mouth closed and open. While mouths were open, the left lower jaw was photographed laterally, the camera being perpendicular to the teeth. In species with villiform dentition, the teeth were found to be angled medially (lingually). Additional images were therefore taken with the camera at approximately 45° to capture the whole tooth length. In species with enlarged lips, the lips were pulled downwards and fixed with a pin to reveal the full length of the teeth. Qualitative observations on the upper jaw dentition patterns were also made. Some species (e.g. *Neoniphon sammara*) have numerous teeth; however, they are so small (generally less than 1 mm) and compact to be almost invisible to the naked eye; for the purpose of this study, they were classified as edentulate as they were too small to measure. Specimens were acquired from commercial suppliers or from donations.

## 2.2. Analysis

Traits based on images were collected using the software ImageJ, and all subsequent data analyses were conducted in the software R [33]. We identified the five largest teeth along the lower jaw and measured these teeth sequentially, from front to back along the jaw. Measured traits were lower jaw length, individual tooth lengths (1–5), distance to jaw tip (1–5), distance between teeth (1–4), largest tooth position from jaw tip, largest tooth width at the base, smallest versus largest tooth length of the five largest teeth, total number of teeth and number of tooth rows (1–5 indicates that the trait was recorded for each of the five teeth). For a detailed description of these traits, see electronic supplementary material, table S1. Throughout our manuscript, the terms largest and smallest teeth are based on tooth length, and therefore also refer to longest and shortest teeth, respectively (given the similarity in tooth shapes). We then converted trait measurements to per cent standard length. To evaluate allometric relationships, we plotted body-standardized variables against body size (SL), and where regressions were significant, calculated residuals. Before transforming values, we produced positive scores by adding a constant to all values (absolute value of the smallest negative residual + 0.1). This treatment eliminates negative residuals (thus allowing transformations) but retains the relationships between scores/trait values.

As morphological variables are not phylogenetically independent, we constructed a phylogenetic tree encompassing all species in our dataset (see electronic supplementary material, figure S1), using the Open Tree of Life [34] and the package '*rotl*' [35]. Tree branch lengths were computed using the Grafen method [36]. We then conducted phylogenetic principal component analyses (PPCA) using the package '*phytools*' [37]. As principal component analysis (PCA) can be sensitive to zeros, and our dataset included zero values describing traits for edentulte (toothless) species, we also analysed our data using a distance-based ordination as opposed to a correlative. We conducted a principal coordinate analysis (PCoA) based on a Gowers distance matrix, using the *vegan* package [38]. After identifying morphotype groupings in these ordinations, we tested the validity of our groupings, by conducting a clustering analysis (simulations = 999, distance method = Euclidean), followed by a similarity profile analysis (SIMPROF) (method = Wards, $\alpha = 0.01$) based on the scores produced from the PPCA (PC1 and PC2), to identify significant clusters using the package '*clustsig*' [39].

After identifying morphotypes based on tooth morphology, we compared these morphotypes with established functional feeding traits. Functional traits were defined as morphological traits for which specific function(s) have been experimentally shown to aid the organism in carrying out a specific task related to feeding. These traits were jaw protrusion [40,41], gape size (vertical oral gape and horizontal oral gape) [26,32], mouth shape (vertical oral gape/horizontal oral gape) [42] and jaw lever ratios [26]. For a detailed description of each function, see electronic supplementary material, table S2. The same treatment applied to morphological tooth traits (evaluating allometry by calculating residuals) was applied to functional traits; however, functional traits were also log10 transformed to minimize the effect of outliers. Following PPCA ordinations to identify distinct functional groups, we analysed each functional trait (same values used for PPCA) (response variable) between morphotypes (explanatory variable) by using phylogenetic least-squares (PGLS) models. PGLS models were conducted to explore the significance (and relationship to morphotypes of each variable individually) of our ordination-based interpretations and were analysed assuming Brownian motion, and using the maximum-likelihood method. Models were conducted using the *nlme* package [43].

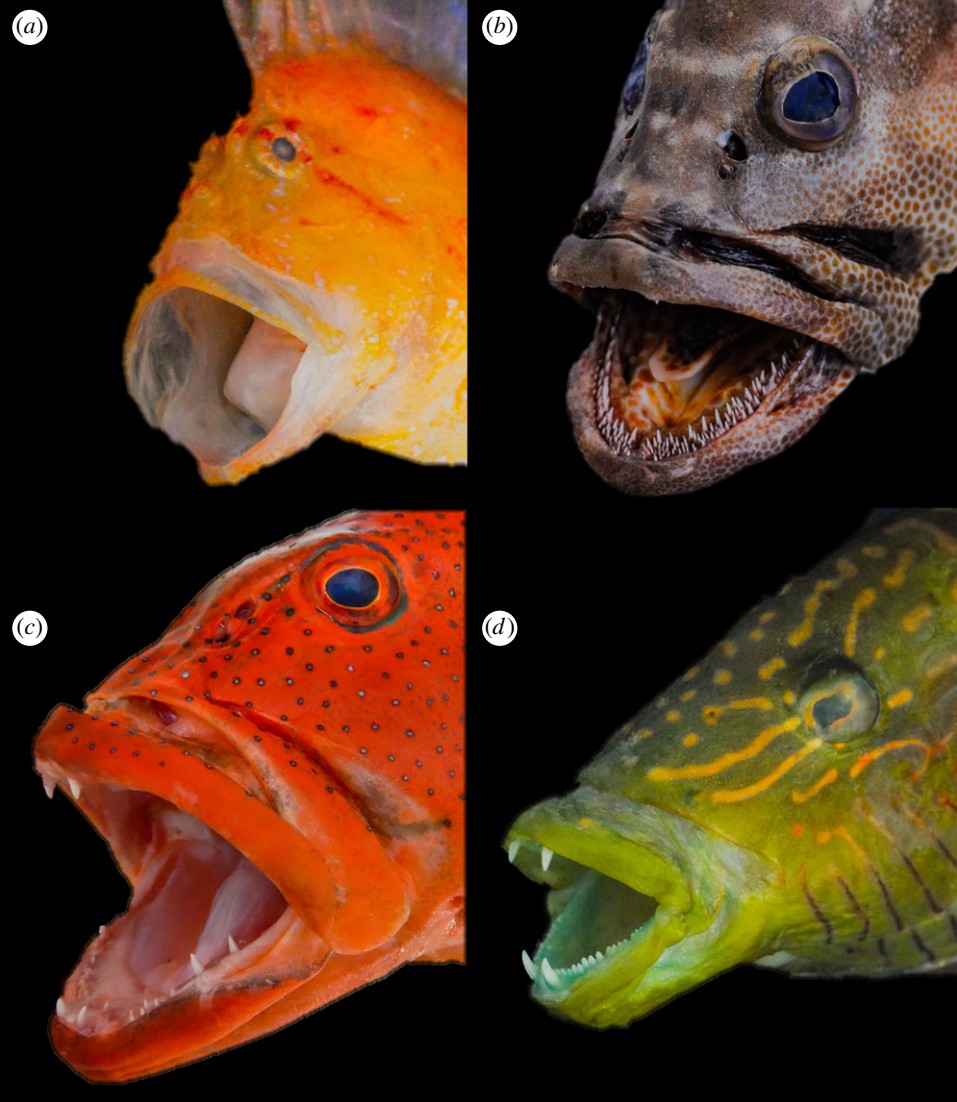

**Figure 1.** Dentition patterns in piscivorous fishes: (*a*) edentulate (*Taenianotus triacanthus*), (*b*) villiform (*Epinephelus polyphekadion*), (*c*) 'back-fanged' macrodont (*Plectropomus leopardus*) and (*d*) 'front-fanged' macrodont (*Oxycheilinus digramma*).

During the initial analysis, we found that some morphological traits did not conform with morphotype divisions. For example, largest tooth position (relative to jaw length) is uninformative for villiform and edentulate fish, as villiform fish have highly homogeneous tooth sizes along their jaw (e.g. figure 1), and edentulate fish teeth are either exceedingly small teeth or absent. We, therefore, undertook a second morphological trait-based analysis where we included only macrodont species (i.e. excluding villiform and edentulate species). In this part of our study, we used a different set of morphological traits which were applicable to macrodont species exclusively. Traits used in the analysis of macrodont species were variance in tooth sizes, smallest versus largest tooth length of the five largest teeth, mean distance between five largest teeth, and largest tooth position. For a detailed description of each trait, see electronic supplementary material, table S1.

## 3. Results

The initial phylogenetic principal component analysis (PPCA), based on tooth-based morphological traits, explained 44.7% (PC1) and 26.6% (PC2), respectively, of the total variation. Three morphotypes, macrodont, villiform and edentulate, are primarily separated along PC1 (figure 2). Villiform species are described by the high abundance of lower jaw teeth (47 in *Cephalopholis microprion* to 96 in

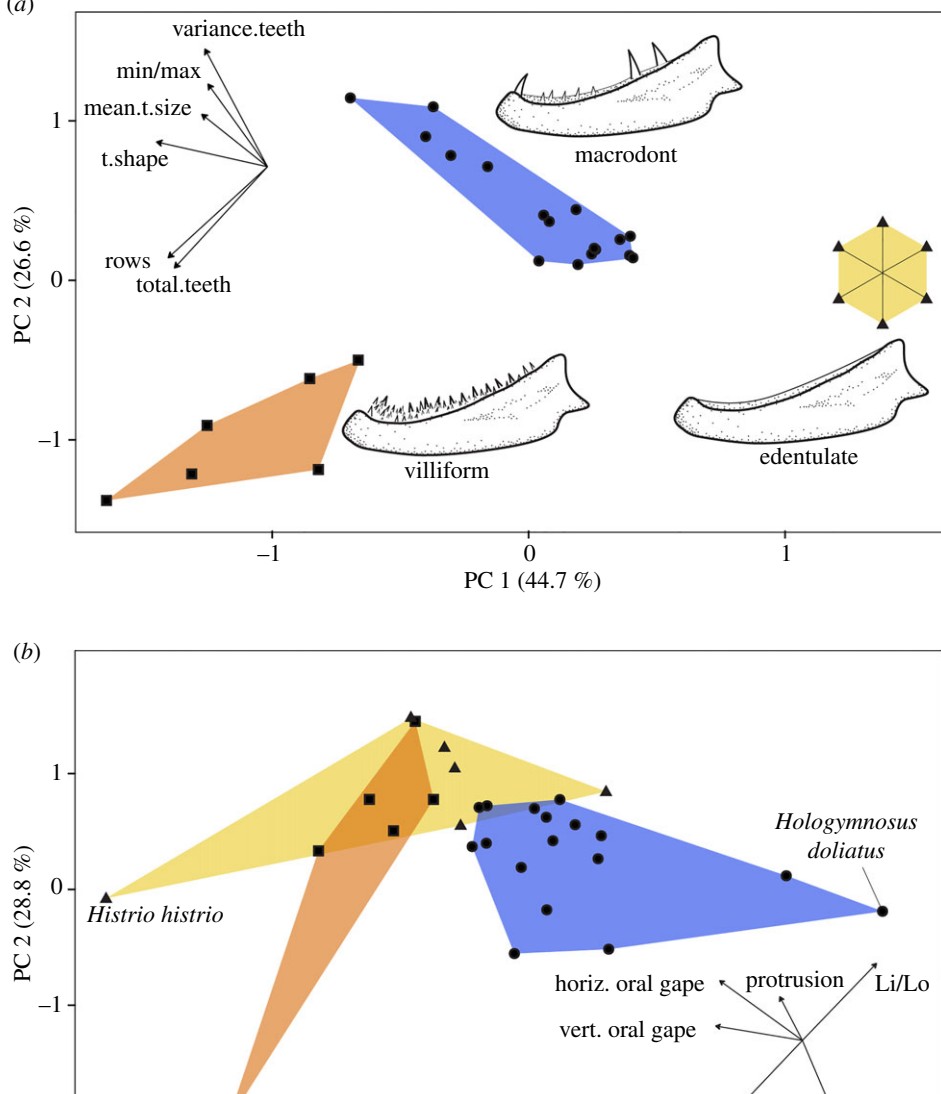

**Figure 2.** PPCA based on (*a*) teeth traits and (*b*) functional feeding traits. Colours/shape scores represent villiform (orange/squares), macrodont (blue/circles) and edentulate (yellow/triangles) species. Lines within the edentulate species polygon are drawn to show that all dots/specimens are in the same location in the ordination. For vector loadings on the principal components, see electronic supplementary material, table S3. For detailed descriptions of traits (tooth and functional), see electronic supplementary material, tables S1 and S2, respectively.

*Epinephelus ongus*) and having three to four tooth rows. Macrodont species are characterized by a higher variance in their teeth sizes (having both large and smaller teeth), with fewer teeth than villiforms (ranging from 4 in *Cheilodipterus* species to 20 in *Hologymnosus annulatus*), usually in a single row. Edentulate species were characterized by having no teeth or teeth which were undetectable with the methods used herein. Our PCoA revealed similar results to our PPCA, suggesting that the zero values of edentulates had minimal effect on our analysis (see electronic supplementary material, figure S2). The clustering and SIMPROF analyses strongly supported our ordination-based groupings (see electronic supplementary material, figure S3). Upper jaw dentition in villiform and macrodont morphotypes was primarily described by a large caniniform tooth on the anteriormost margin of the premaxilla (usually smaller in species with villiform dentition), followed posteriorly by smaller similarly shaped teeth (see figure 6).

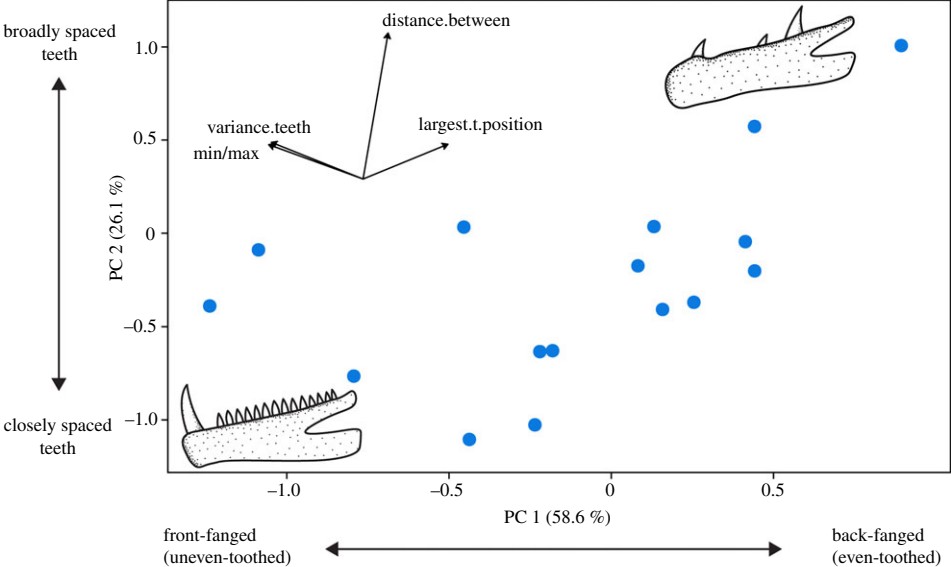

**Figure 3.** PPCA of macrodont piscivores. For vector loadings on principal components, see electronic supplementary material, table S3. For detailed description of scores, see electronic supplementary material, figure S6.

In contrast with the morphological trait PPCA, the functional trait PPCA revealed more overlap between tooth morphotypes, especially between edentulate and villiform morphotypes. These two morphotypes were mostly separated from macrodonts by having larger gape sizes (figure 2). Macrodonts were characterized by smaller gape sizes and higher Lo/Li values (velocity advantage) (figure 2). Both gape sizes and Lo/Li traits were significant in the PGLS models (electronic supplementary material, table S4). Mouth shape (ratio of vertical oral gape/horizontal oral gape) is vertically oval in macrodont species, whereas edentulate and villiform species were characterized by more rounded mouths; this was, however, not significant in PGLS models (electronic supplementary material, table S4). Jaw protrusion appears to be mostly associated with edentulate and villiform morphotypes; this was, however, not significant in PGLS models (electronic supplementary material, table S4). One species, *Saurida argentea*, does not fit the functional pattern of the rest of villiforms, as it is characterized by a high-velocity advantage jaw (high Lo/Li ratio), but no protrusion (figure 2b). In essence, our results show edentulate and villiform morphotypes to be characterized by larger gape sizes and lower velocity advantage in jaw closing, whereas macrodonts were characterized by smaller gape sizes and higher velocity advantage in jaw closing. It appears that while there may be three-tooth morphotype groups, functionally, there are probably only two groups, macrodonts versus villiform/edentulate.

When macrodonts were analysed exclusively, *Cheilodipterus macrodon* was an outlier and was therefore removed from the analysis (for an ordination including this outlier, see electronic supplementary material, figure S4). Excluding *Cheilodipterus*, our macrodont-based PPCA explained 58.6% (PC1) and 26.1% (PC2), respectively, of the total variation (figure 3). PC1 is mostly associated with 'variance in tooth sizes' and 'min/max ratio', and 'position of largest tooth'. This axis suggests a continuum between species with one large tooth (sometimes two teeth) located anteriorly on the jaw followed posteriorly by smaller teeth versus species with similar-sized teeth (note this refers to the five largest teeth, not all teeth), where the largest tooth is located posteriorly on the jaw (occasionally a similar-sized caniniform tooth is present in the anteriormost point of their jaw). Extremes of this continuum are hereby termed 'front-fanged' and 'back-fanged', respectively. PC2 is mostly associated with 'distance between teeth', indicating an axis of variation between 'broadly spaced' versus 'closely spaced' teeth. In essence, our results suggest a continuum between 'front-fanged' dentition types which have a large anterior tooth, with teeth being unevenly sized and tightly spaced versus 'back-fanged' dentition types which have a large posterior tooth, with broad tooth spacing, and even tooth sizes. If tooth force potential (based on lever-ratio mechanics) is calculated for anteriormost versus posteriormost caniniform teeth in back-fanged dentition types, posteriormost teeth were found to have an average 42.1% increase in force (figure 4). These potential morphotypes appear to be independent of body size and jaw length, as both morphotypes were distributed along the entire range of our sampled body sizes and jaw lengths (electronic supplementary material, figure S5).

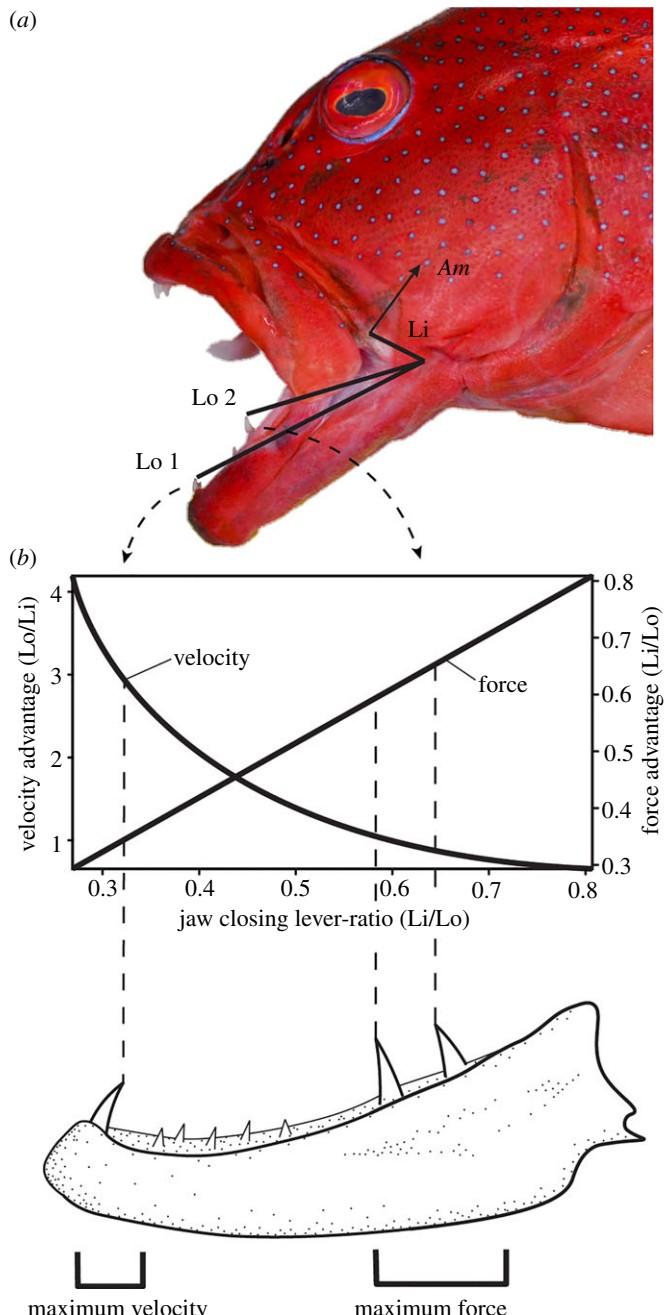

**Figure 4.** Lever-ratio biomechanics for teeth of a back-fanged macrodont piscivore (*P. leopardus*): (*a*) change in force between anteriormost versus posteriormost teeth. Li, in-lever; Lo1, out-lever to the anteriormost tooth; Lo2, out-lever to the posteriormost largest tooth; Am, *adductor mandibulae* muscle. (*b*) Relationship between velocity advantage and force advantage when calculating lever ratios (modified after [29]) and the functional ramifications of this principal for anteriorly versus posteriorly positioned canines.

## 4. Discussion

Our analyses identified three major tooth-based morphotypes in piscivorous fishes: edentulate, villiform and macrodonts. We found that tooth shape, relative tooth size and the number of teeth (along with tooth rows) were the primary distinguishing features of these morphotypes (figure 2*a*). Also, when analysed in a context of functional feeding traits, edentulate and villiform morphotypes were found to be overlapping, whereas macrodonts were distinct (figure 2*b*). Edentulate and villiform fishes were characterized by larger gape sizes and lower velocity advantage in jaw closing, and to a lesser extent, more rounded mouth openings. Macrodonts were characterized by smaller gape sizes and higher

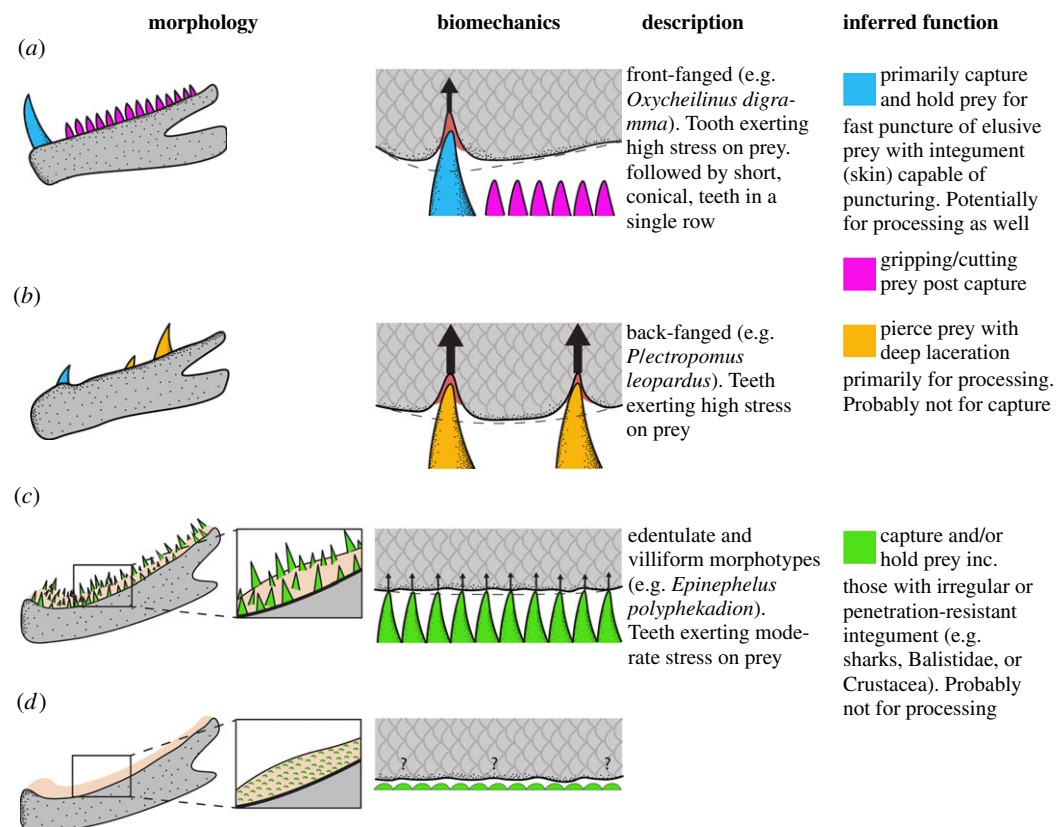

|  | morphology | biomechanics | description | inferred function |

**Figure 5.** Dentition morphotypes displaying individual tooth morphology, biomechanical properties, description and inferred function based on tooth size, position along the jaw, number of teeth, number of teeth rows and distance between teeth: (a) front-fanged macrodont, (b) back-fanged macrodont, (c) villiform and (d) edentulate.

velocity advantage, and to a lesser extent, more oval-shaped mouths (figure 2b). When macrodonts were analysed exclusively, we found a distinct axis of variation, which may reflect functional divergences in the oral teeth of fishes and other homodont lower vertebrates. We suggest that tooth function for some lower vertebrates might differ not based on tooth shape but solely by position along the jaw. In other words, even if organisms are homodont (like the vast majority of vertebrate species), functional diversification is still possible.

As the functional traits used in our study are key to the prey capture and/or post-capture processing of prey, it is likely that edentulate and villiform fishes will display similar feeding behaviours that are quite distinct when compared with macrodonts. However, for these behaviours to be displayed, and quantified accurately, these organisms may need to be tested in a maximal performance-based context [28,32,44]. For example, Reimchen [45] showed that capturing and processing behaviour for a predatory fish was random for small-sized prey, but shifted to head-first processing when predators were fed prey with body diameter over half their gape. Mihalitsis & Bellwood [32] likewise found *Cephalopholis urodeta*, a piscivore with villiform dentition and a relatively large gape, captured prey head-first, whereas *Paracirrhites forsteri*, a macrodont with a smaller gape, captured prey mid-body or tail first. Based on the observed morphologies and behaviour, we suggest that edentulate and villiform species, with larger gape sizes, might be more efficient in 'engulfing' their prey through ambush predation, whereas macrodonts, with smaller gape sizes but larger teeth, might be more efficient at 'grabbing' their prey after a short-distance lunge and/or longer pursuit. This axis of variation may also reflect varying contributions from suction versus ram in engulfing versus grabbing species [41,46,47].

After prey capture, Mihalitsis & Bellwood [32] found that *P. forsteri* conducted a series of head-shaking movements when processing prey, potentially to slash/lacerate prey by using their teeth. This feeding behaviour of head shaking is similar to that seen in non-teleostean fish groups, e.g. chondrichthyans [48], especially when feeding on prey too large to swallow whole [49]. Interestingly, this behaviour is also observed in other vertebrates (e.g. lizards) [50–52], with similar tooth morphotypes (figure 7).

When analysing macrodonts exclusively, we found the main axis of variation (PC1) displayed a continuum, with the extremes being 'front-fanged' and 'back-fanged' species. Teeth used to penetrate

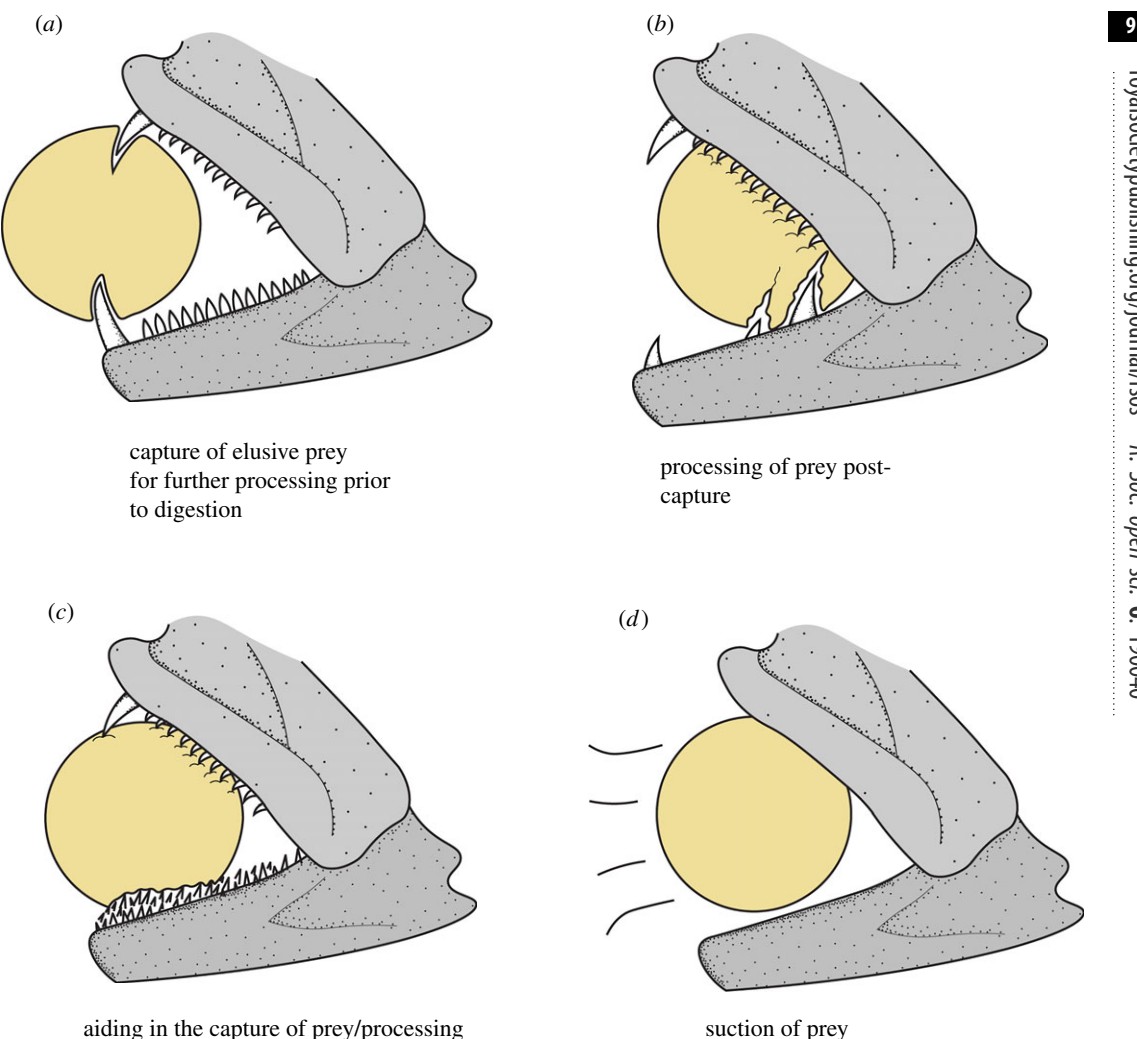

**Figure 6.** Full dentition morphotypes (both upper and lower jaws) displaying inferred functional capabilities based on biomechanical properties: (*a*) front-fanged macrodont, (*b*) back-fanged macrodont, (*c*) villiform and (*d*) edentulate. Note the absence of the back-fanged dentition in upper jaws.

prey are strongly linked to the biomechanical property of stress [53,54], that is, the force applied to an object relative to the area over which it is applied ($\sigma$ = force/area, SI = N m$^{-2}$) [15,55]. Having a single large caniniform tooth followed (or surrounded) by small teeth maximizes the stress the large tooth will exert on prey tissues, just like having multiple similar-sized teeth but positioned further apart (figure 5). In villiform dentition, similarly shaped teeth in large numbers are likely to act like a 'bed of nails' which may be able to grip rather than puncture (figure 5). This observation highlights the need to look past single-tooth morphology alone and integrate full dentition-based studies when elucidating the life history of organisms.

Furthermore, we suggest that 'back-fanged' dentition patterns may have key functional implications based on lever-ratio biomechanics. By having a large caniniform tooth posteriorly in their jaw, these species gain, on average, a 42% force advantage when compared with a same-sized tooth positioned at the anteriormost point of the jaw (figure 4) (see also [15]). This value mirrors differences reported in anterior versus posterior jaw bite pressure, calculated in [15] for king mackerel (*Scomberomorus cavalla*). This increase in force advantage could provide the predator with the force required to deeply pierce prey. We suggest that back-fanged morphotypes could be exhibiting a form of functional decoupling, with the anteriormost canines (higher speed/less force) being used for grabbing prey, whereas posteriorly positioned canines (lower speed/higher force) are used for post-capture processing (figure 4), such as deeply piercing and/or lacerating prey, especially when using the slashing behaviour described above. This is further highlighted by the lack of back-fanged tooth distributions on the upper jaw (figure 6).

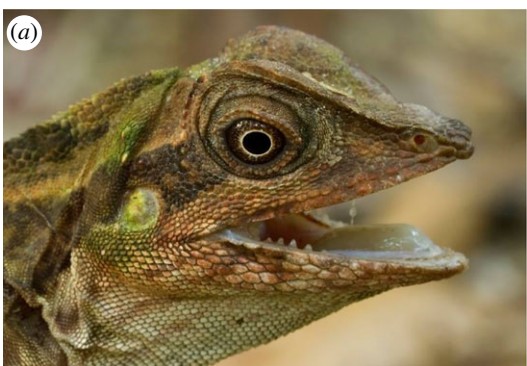
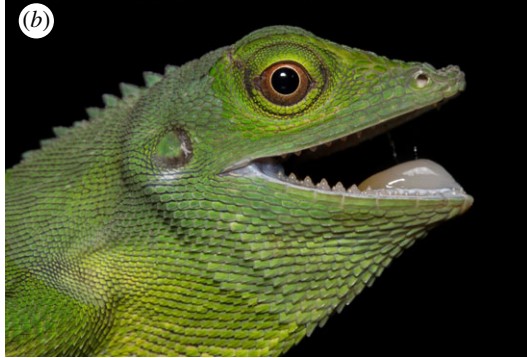

**Figure 7.** Back-fanged dentition types observed in non-fish vertebrate groups, such as lizards: (*a*) *Gonocephalus grandis* and (*b*) *Bronchocela cristatella*. Photo credits, respectively: Henrik Bringsøe and Elliot Budd.

Functional decoupling has long been suggested to provide an evolutionary advantage, for example, in the fused pharyngeal jaws of cichlids (i.e. pharyngognathy) [56–59]. Pharyngeal jaws in piscivorous cichlids have been identified multiple times as a means of processing/lacerating prey [60–62], suggesting that piscivorous cichlid species might not use their oral teeth for processing, only capturing. By contrast, the influence of pharyngeal jaws on prey processing in non-pharyngognath piscivores has been suggested to be negligible [62]. This could suggest that while pharyngognath piscivores may capture prey using their oral teeth and process it using their pharyngeal jaws, non-pharyngognath piscivores may both capture and process prey with their oral jaws. In this regard, we note that back-fanged species do not seem to be represented in pharyngognath piscivorous cichlids (Cichlidae) [63,64], offering support for the suggestion that back-fanged oral teeth in non-pharyngognath species could have a similar function to that of the pharyngeal jaw teeth of pharyngognaths (i.e. lacerating/processing prey). Based on previous observations, and the results from our study, we suggest that some form of functional decoupling could be present within the oral jaws of fishes, and not just between oral versus pharyngeal jaw systems.

If back-fanged species represent a functional decoupling of the oral teeth, separating fast grabbing anterior teeth from slower but deeply penetrating posterior teeth, a longer lower jaw would maximize both the velocity advantage of the anterior tooth and the force advantage of the posterior tooth (relative to the anterior tooth). Interestingly, lower jaw elongation has arisen on multiple occasions and has been widely associated with increased piscivory [62,65,66]. It has been suggested that the mechanistic function underlying jaw elongation is an increase in gape size and creating a larger contact area between predator and prey for prey manipulation [14,67]. Here, we suggest that the mechanistic function of jaw elongation may be to facilitate separation of front fangs, for capture, from back-fangs, with increased pressure/stress output, for prey manipulation and processing (figure 5).

Overall, we provide a quantitative framework for identifying dentition morphotypes in lower vertebrates, especially piscivorous fishes and provide a putative functional interpretation of these distinct morphotypes. We identify three distinct dentition morphotypes (edentulate/villiform/macrodont) that appear to be encompassed by just two functional groups, broadly classified as 'engulfers' versus 'grabbers'. Also, within macrodonts, we identify a continuum between front-fanged and back-fanged species and explore the functional implications separating teeth involved in procurement (grabbing) versus processing (laceration). We highlight the potential for functional decoupling in fish teeth, based not on the shape of the tooth, but their relative position along the jaw.

Ethics. All measurements were carried out at James Cook University (JCU), Queensland, Australia, in accordance with the JCU Animal Ethics Committee (A2181 approved 08/5/15, and A2529 approved 1/6/18).

Data accessibility. All data used in this study are available as electronic supplementary material (see file 'Raw Data'). For code used in R software, see file 'R software code' in the electronic supplementary material.

Authors' contributions. M.M. and D.B. conceived and designed the study, M.M. carried out the data collection, analyses and prepared the initial manuscript and figures, M.M. and D.B. prepared the final manuscript. Both the authors gave final approval for publication.

Competing interests. We have no competing interests.

Funding. This study was funded by the Australian Research Council (DRB: grant no. CE140100020. URL: http://www.arc.gov.au/).

Acknowledgements. We thank Hillcoat K., Thompson A. for fish specimens and Goatley C., Hemingson C., Huertas V., Morais R., Siquera A., Streit R. and Tebbett S. for insightful discussions. We also thank Henrik Bringsøe and Elliot Budd for lizard photographs.

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
