## [Reviewer comments · Royal Society Open Science]

Review History

RSOS-190040.R0 (Original submission)

Review form: Reviewer 1

Is the manuscript scientifically sound in its present form?

No

Are the interpretations and conclusions justified by the results?

Yes

Is the language acceptable?

Yes

Is it clear how to access all supporting data?

No

Do you have any ethical concerns with this paper?

No

Have you any concerns about statistical analyses in this paper?

No

Recommendation?

Major revision is needed (please make suggestions in comments)

Comments to the Author(s)

This is an interesting manuscript focused on function of "homodont" teleost fishes. Homodont of course is a term used in reference to a dentition comprised of a single tooth type (vs. heterodont).

The study is interesting and overall I am OK with much of the manuscript. However, I do have several issues with the introduction and also how the authors define homodont (which has implications for some of the authors arguments). I list my major issues below and then end the review with a few minor issues for the authors to take care of (minimal effort on their part).

Major issues:

1. How do the authors define homodont? A fish with small conical teeth all of the same size (villiform for example) would be a text book homodont in my opinion. But the authors focus on at least two species in which the teeth are of very different sizes (see there fig. 2c/d). This includes fishes with larger "canine" teeth either at the front or the rear of the jaws combined with smaller conical teeth. In order for these species to be homodont the canine teeth and the conical teeth would have to be the same type. Do the authors really consider these larger canines to be the same as the smaller conical teeth? One way to convince the reader of this would be to show that the canines were simply scaled up replicas of the smaller conical teeth (or visa versa). If they can not show this then I think this is a major flaw. These fishes would be heterodonts and not homodonts, which would require some restructuring of the introduction, and maybe discussion.

2. The introduction is set up to convince the reader that there are almost no studies on the teeth of fishes, and those that are available are only descriptive. I think this is really not the case and leads the reader a stray. For example, if one was to type "shark and teeth" or "teleost and teeth" into a search engine such as google scholar the number of returns would be >58,000 and >11,000, respectively. Of course, these numbers would include the descriptive studies that they authors make reference too but many, many "non-descriptive" studies on the teeth of fishes would also be included. In my opinion, the authors give these studies short shrift. I understand that the authors are trying to sale the study as a "first" (and in several aspects it is) but it is a little sneaky (maybe even deceitful) to try to convince the reader that there are almost no functional studies on the teeth of fishes. If one was to conduct a decent review, I am sure that there would be just as many studies on the teeth of fishes as there are on the teeth of mammals. My major recommendation is that the authors tone down the introduction.

Minor comments:

P3, L73: chondichthyans should be chondrichthyans

P4, L84: First mention of piscivorous. Rather abrupt....

P10, L "a villiform with a relatively large gape," does not make sense. The fish is not a villiform, it has a villiform dentition. Please revise.

P10, L242-244: "(see Supplemental Fig. 6 for dentition of Varanus monitor lizard, a species also appearing to be `back-fanged`, and other fish orders (e.g. Characiformes)." There is a problem with the brackets here. There are not enough or too many. Please correct. Also, seems a little out of context to provide a S figure of a lizard? The point could be made without the figure. Also, I can not even see the S figures.

Supplementary material was not made available to the reviewer and would have been helpful.

Review form: Reviewer 2

Is the manuscript scientifically sound in its present form?

Yes

Are the interpretations and conclusions justified by the results?

Yes

Is the language acceptable?

Yes

Is it clear how to access all supporting data?

Yes

Do you have any ethical concerns with this paper?

No

Have you any concerns about statistical analyses in this paper?

No

Recommendation?

Accept with minor revision (please list in comments)

Comments to the Author(s)

Good introduction. On line 81-82 you point out that homodont teeth have similar shapes but there might be other aspects of morphology that influence tooth function – at this point I would suggest you just elaborate on this a bit and add some verbiage to the effect “... including tooth size, position and occlusion patterns”.

On a separate note, the problem with the homodont versus heterodont dichotomy is that sometimes there is a grey zone with what is termed homodont. It is clear that teleosts such as the drums have a heterodonty dentition with more molariform teeth posteriorly, but some other fishes have very recurved and pointy teeth anteriorly, and more triangular teeth throughout the jaw or posteriorly. Is this homodont? Heterodontus the shark is clearly heterodonty- but is the mako shark homodont or heterodont? Maybe you should very briefly mention in the intro that sometimes the terms are a bit vague and there are fishes that don't quite fit the dichotomy.

In the methods line 96 do you mean vertical and horizontal oral gape distance? You just say “gapes”. It could be gape angle or distance. Clarify

Line 102 – what do you mean by “inwards”. Do you also mean the whole fish head was tilted at an angle to envision the teeth – and how was it tilted? You can't just tilt the teeth. This is all unclear.

Did you sometimes cut away a bit of labia (gums) to expose the total teeth? Sometimes the “gum” hides the base of the teeth and you can't see the total length nor the base. We do this sometimes.

Line 105 – I do disagree that you term fish with very small teeth as edentulate. To me you need another name for this. Cypriniforms are edentulate. Some fish have very small villiform teeth; you could even call them cardiform, but NOT edentulate.

In the Analysis section here are some comments: You need to change the terminology largest to longest (throughout) if that is truly what you mean. Smallest should also be changed to shortest if that is what you mean. Large and small refers to size, which is different from long and short.

Line 113 largest tooth width “at the base”

Line 111 Measured units were : lower jaw length...

What about upper jaw teeth? If they mirror lower jaw teeth say so. Some fishes have extremely large upper jaw teeth and your analysis does not account for that (gempylids).

Line 138 – “vertical gape height” you say gape size which is not specific and on that line you say “mouth shape” what does that mean, be specific. Do you mean circular, notched, laterally occluded?

Line 150 and other similar sections- you say largest tooth position- I think you mean longest tooth position. If you mean largest then you must define what constitutes “largest” – is that base width and length, height also, what? Do a global search for largest, large, small, smallest and replace with the correct terms. On figure 1 I now see horizontal oral gape and vertical oral gape. This needs to be spelled out in the analysis sections. Was gape size measured at the anterior tips of the upper and lower jaws? Some researchers have measured it at the notch part of the gape.

Results

Line 168- remove qualifiers like “highly” – just say similar. The problem with this is, how “high” is highly?

Line 179 – oval mouths- do you mean notched?

What about protrusion characteristics? Does the macrodont tooth type have generally less protrusion than the villiform/edentulate forms? That wasn’t clear in the text.

Line 194- located anteriorly ON the jaw- it cant be anterior to the jaw, nor posterior to the jaw, so correct those sentences.

Line 201 –unevenly sized

I really liked how you summarized each type with a clear description following the details- nice job.

Discussion

Lines 234-236- are you referring to suction versus ram capture? Engulfing and grabbing are not terms we usually use in the prey capture literature. What is also important here is the difference between getting the prey to the mouth, and then processing or holding the prey in the jaws. You need to be clearer regarding this difference. Edentulous species or those with villiform/cardiform teeth may use some degree of suction to capture the prey and the fine teeth to hold it (hence the rounder mouth aperture); macrodents tend to use more ram (barracuda), grasping and cutting the prey with the enlarged teeth. However, there is more than one way to skin a cat (many to one mapping-Wainwright) and wahoo have relatively small teeth, all about the same size, and use incredible ram. Then some fish such as grouper will use more ram or suction (with relatively small teeth) depending on where the prey are (Collins & Motta 2017 A kinematic investigation into the feeding behavior of the Goliath Grouper *Epinephelus itajara*. *Environ Biol Fish* 100: 309-323.)

The section ending on line 25 brings to mind the study on tooth pressure by macrodont piscivores- Ferguson et al 2015 Feeding performance of king mackerel *J Exp Zool* 323A:399-413. These extremely sharp teeth, especially at the rear of the jaw, exert extreme pressure (force/unit area) on the prey. You might discuss that. This could also fit after the section ending on line 265. I believe there is one other fish study that correctly calculated bite pressure – it may be Westneat or Anderson. Most confuse force and pressure- and all the pressure stuff on line is crap! Really good point on lines 282-286 – reminds me of the diminutive Pike Killifish *Belonesox belizanus* - a really weird killifish- look at the pictures.

Rather than the ending paragraph I would suggest a summary paragraph – a lot of the time the reader zeros in on the final summary of the findings and main conclusions. Yours is a bit too general and the punch line is lost.

Decision letter (RSOS-190040.R0)

21-May-2019

Dear Mr Mihalitsis,

The editors assigned to your paper ("Functional implications of dentition-based morphotypes in piscivorous fishes") have now received comments from reviewers. We would like you to revise your paper in accordance with the referee and Associate Editor suggestions which can be found below (not including confidential reports to the Editor). Please note this decision does not guarantee eventual acceptance.

Please submit a copy of your revised paper before 13-Jun-2019. Please note that the revision deadline will expire at 00.00am on this date. If we do not hear from you within this time then it will be assumed that the paper has been withdrawn. In exceptional circumstances, extensions may be possible if agreed with the Editorial Office in advance. We do not allow multiple rounds of revision so we urge you to make every effort to fully address all of the comments at this stage. If deemed necessary by the Editors, your manuscript will be sent back to one or more of the original reviewers for assessment. If the original reviewers are not available, we may invite new reviewers.

- Data accessibility

If you wish to submit your supporting data or code to Dryad (<http://datadryad.org/>), or modify your current submission to dryad, please use the following link:
<http://datadryad.org/submit?journalID=RSOS&manu=RSOS-190040>

- **Competing interests**

- **Authors' contributions**

- **Acknowledgements**

- **Funding statement**

on behalf of Professor Emily Standen (Associate Editor) and Kevin Padian (Subject Editor)
openscience@royalsociety.org

Associate Editor's comments (Professor Emily Standen):

Associate Editor: 1

Comments to the Author:

Dear Dr. Mihalitsis,

We have now received both reviews of your manuscript entitled the Functional implications of dentition-based morphotypes in piscivorous fishes. In general the reviews are very positive and

find the work interesting and valuable. Both reviewers do suggest clarifying the definition of homodonty in your paper as well as being more specific regarding the lack of literature on fish dentition. As one reviewer points out, there is a large body of literature on fish teeth and this should be recognized.

I look forward to seeing your careful address for the reviewers comments.

Sincerely,

Emily Standen

Associate Editor: 2

Comments to the Author:

Dear Michalis Mihalitsis,

Thank you for your submission of the manuscript entitled Functional implications of dentition-based morphotypes in piscivorous fishes. I find this an interesting topic and see value in broadening our ability to assess ecomorphology of basal vertebrates using teeth to compare with the tools that have been developed for mammalian teeth.

I am in agreement with the reviewer who suggests two main points that need to be addressed to increase the clarity and impact of your work. First, reconsider the use of the word homodonty, or at the very least expand your definition to help avoid confusion for readers, and second, incorporate more of the existing literature on fish teeth.

At this point we must reject your paper but we encourage you to fully address the reviewers comments and resubmit the paper for further review.

Emily Standen

Editor comments:

Thanks for revising. As you can see, the first reviewer still has major concerns with the paper, especially with definitions and analyses. The second reviewer additionally points out that there is a considerable literature on this topic that is not acknowledged. This is a very important issue for us.

Please address all of the reviewers' comments in your revision. Best wishes.

Comments to Author:

Reviewers' Comments to Author:

Reviewer: 1

Comments to the Author(s)

This is an interesting manuscript focused on function of "homodont" teleost fishes. Homodont of course is a term used in reference to a dentition comprised of a single tooth type (vs. heterodont).

The study is interesting and overall I am OK with much of the manuscript. However, I do have several issues with the introduction and also how the authors define homodont (which has implications for some of the authors arguments). I list my major issues below and then end the review with a few minor issues for the authors to take care of (minimal effort on their part).

Major issues:

1. How do the authors define homodont? A fish with small conical teeth all of the same size (villiform for example) would be a text book homodont in my opinion. But the authors focus on at least two species in which the teeth are of very different sizes (see there fig. 2c/d). This includes fishes with larger "canine" teeth either at the front or the rear of the jaws combined with smaller conical teeth. In order for these species to be homodont the canine teeth and the conical teeth would have to be the same type. Do the authors really consider these larger canines to be the same as the smaller conical teeth? One way to convince the reader of this would be to show that the canines were simply scaled up replicas of the smaller conical teeth (or visa versa). If they can not show this then I think this is a major flaw. These fishes would be heterodonts and not homodonts, which would require some restructuring of the introduction, and maybe discussion.

2. The introduction is set up to convince the reader that there are almost no studies on the teeth of fishes, and those that are available are only descriptive. I think this is really not the case and leads the reader a stray. For example, if one was to type "shark and teeth" or "teleost and teeth" into a search engine such as google scholar the number of returns would be >58,000 and >11,000, respectively. Of course, these numbers would include the descriptive studies that they authors make reference too but many, many "non-descriptive" studies on the teeth of fishes would also be included. In my opinion, the authors give these studies short shrift. I understand that the authors are trying to sale the study as a "first" (and in several aspects it is) but it is a little sneaky (maybe even deceitful) to try to convince the reader that there are almost no functional studies on the teeth of fishes. If one was to conduct a decent review, I am sure that there would be just as many studies on the teeth of fishes as there are on the teeth of mammals. My major recommendation is that the authors tone down the introduction.

Minor comments:

P3, L73: chondichthyans should be chondrichthyans

P4, L84: First mention of piscivorous. Rather abrupt....

P10, L "a villiform with a relatively large gape," does not make sense. The fish is not a villiform, it has a villiform dentition. Please revise.

P10, L242-244: "(see Supplemental Fig. 6 for dentition of *Varanus* monitor lizard, a species also appearing to be `back-fanged`, and other fish orders (e.g. Characiformes)." There is a problem with the brackets here. There are not enough or too many. Please correct. Also, seems a little out of context to provide a S figure of a lizard? The point could be made without the figure. Also, I can not even see the S figures.

Supplementary material was not made available to the reviewer and would have been helpful.

Reviewer: 2

Comments to the Author(s)

Good introduction. On line 81-82 you point out that homodont teeth have similar shapes but there might be other aspects of morphology that influence tooth function – at this point I would suggest you just elaborate on this a bit and add some verbiage to the effect "... including tooth size, position and occlusion patterns".

On a separate note, the problem with the homodont versus heterodont dichotomy is that sometimes there is a grey zone with what is termed homodont. It is clear that teleosts such as the drums have a heterodonty dentition with more molariform teeth posteriorly, but some other fishes have very recurved and pointy teeth anteriorly, and more triangular teeth throughout the jaw or posteriorly. Is this homodont? Heterodontus the shark is clearly heterodonty- but is the

mako shark homodont or heterodont? Maybe you should very briefly mention in the intro that sometimes the terms are a bit vague and there are fishes that don't quite fit the dichotomy. In the methods line 96 do you mean vertical and horizontal oral gape distance? You just say "gapes". It could be gape angle or distance. Clarify

Line 102 – what do you mean by "inwards". Do you also mean the whole fish head was tilted at an angle to envision the teeth – and how was it tilted? You can't just tilt the teeth. This is all unclear.

Did you sometimes cut away a bit of labia (gums) to expose the total teeth? Sometimes the "gum" hides the base of the teeth and you can't see the total length nor the base. We do this sometimes. Line 105 – I do disagree that you term fish with very small teeth as edentulate. To me you need another name for this. Cypriniforms are edentulate. Some fish have very small villiform teeth; you could even call them cardiform, but NOT edentulate.

In the Analysis section here are some comments: You need to change the terminology largest to longest (throughout) if that is truly what you mean. Smallest should also be changed to shortest if that is what you mean. Large and small refers to size, which is different from long and short.

Line 113 largest tooth width "at the base"

Line 111 Measured units were : lower jaw length...

What about upper jaw teeth? If they mirror lower jaw teeth say so. Some fishes have extremely large upper jaw teeth and your analysis does not account for that (gempylids).

Line 138 – "vertical gape height" you say gape size which is not specific and on that line you say "mouth shape" what does that mean, be specific. Do you mean circular, notched, laterally occluded?

Line 150 and other similar sections- you say largest tooth position- I think you mean longest tooth position. If you mean largest then you must define what constitutes "largest" – is that base width and length, height also, what? Do a global search for largest, large, small, smallest and replace with the correct terms. On figure 1 I now see horizontal oral gape and vertical oral gape. This needs to be spelled out in the analysis sections. Was gape size measured at the anterior tips of the upper and lower jaws? Some researchers have measured it at the notch part of the gape.

Results

Line 168- remove qualifiers like "highly" – just say similar. The problem with this is, how "high" is highly?

Line 179 – oval mouths- do you mean notched?

What about protrusion characteristics? Does the macrodont tooth type have generally less protrusion than the villiform/edentulate forms? That wasn't clear in the text.

Line 194- located anteriorly ON the jaw- it cant be anterior to the jaw, nor posterior to the jaw, so correct those sentences.

Line 201 –unevenly sized

I really liked how you summarized each type with a clear description following the details- nice job.

Discussion

Lines 234-236- are you referring to suction versus ram capture? Engulfing and grabbing are not terms we usually use in the prey capture literature. What is also important here is the difference between getting the prey to the mouth, and then processing or holding the prey in the jaws. You need to be clearer regarding this difference. Edentulous species or those with villiform/cardiform teeth may use some degree of suction to capture the prey and the fine teeth to hold it (hence the rounder mouth aperture); macrodents tend to use more ram (barracuda), grasping and cutting the prey with the enlarged teeth. However, there is more than one way to skin a cat (many to one mapping-Wainwright) and wahoo have relatively small teeth, all about the same size, and use incredible ram. Then some fish such as grouper will use more ram or suction (with relatively small teeth) depending on where the prey are (Collins & Motta 2017 A kinematic investigation into the feeding behavior of the Goliath Grouper *Epinephelus itajara*. *Environ Biol Fish* 100: 309-323.)

The section ending on line 25 brings to mind the study on tooth pressure by macrodont piscivores- Ferguson et al 2015 Feeding performance of king mackerel J Exp Zool 323A:399-413. These extremely sharp teeth, especially at the rear of the jaw, exert extreme pressure (force/unit area) on the prey. You might discuss that. This could also fit after the section ending on line 265. I believe there is one other fish study that correctly calculated bite pressure – it may be Westneat or Anderson. Most confuse force and pressure- and all the pressure stuff on line is crap! Really good point on lines 282-286 – reminds me of the diminutive Pike Killifish *Belonesox belizanus* - a really weird killifish- look at the pictures. Rather than the ending paragraph I would suggest a summary paragraph – a lot of the time the reader zeros in on the final summary of the findings and main conclusions. Yours is a bit too general and the punch line is lost.

Author's Response to Decision Letter for (RSOS-190040.R0)

See Appendix A.

RSOS-190040.R1 (Revision)

Review form: Reviewer 1

Is the manuscript scientifically sound in its present form?

Yes

Are the interpretations and conclusions justified by the results?

Yes

Is the language acceptable?

Yes

Do you have any ethical concerns with this paper?

Yes

Have you any concerns about statistical analyses in this paper?

Yes

Recommendation?

Accept as is

Comments to the Author(s)

The authors have tried to appease the reviewers from the last round, including myself. I think that they have done a good job. I do not know why they could not cite a few more fish papers on heterodonty in bony fishes, including those by younger authors (e.g., K. Bemis, K. Conway).....that would help the whole community as a whole, give them a hand when possible!

I look forward to seeing the final version.

Review form: Reviewer 2

Is the manuscript scientifically sound in its present form?

Yes

Are the interpretations and conclusions justified by the results?

Yes

Is the language acceptable?

Yes

Do you have any ethical concerns with this paper?

No

Have you any concerns about statistical analyses in this paper?

No

Recommendation?

Accept with minor revision (please list in comments)

Comments to the Author(s)

Intro line 61- canines is misspelled

Line 81-82 establishing a functional link between a certain anatomical features and how (this) help(s) the organism perform a specific task – missing word

Methods- first line- remove comma after measured

Line 111- still have a problem with the term medially- teeth were angled lingually or labially- is that what you mean? I don't know what medially refers to in this case.

End of methods- "In this part of our study, we used a different set of morphological traits which were applicable to macrodont species exclusively. Traits used here were: variance in tooth sizes, smallest vs. largest tooth length of the five largest teeth, mean distance between five largest teeth, and largest tooth position. For a detailed description of each trait see Supplemental Table 1". Are these in addition to the forementioned morphological traits? If so, say so. Most readers won't go to the supplemental table.

Results line 194- in macrodont needs a space between words

In the discussion on lines 305-307

"If back fanged species represent a functional decoupling of the oral teeth, separating fast grabbing anterior teeth from slower but deeply penetrating posterior teeth, a longer lower jaw would maximize both the velocity advantage of the anterior tooth, and the (RELATIVE) force advantage of the posterior tooth."

The rear force advantage would be RELATIVE to the anterior velocity advantage- I don't think it would be an absolute. There would just be a relatively larger difference between front and back. If the adductor muscle is the same size developing the same force between two such fishes, and the rear tooth is the same distance from the jaw joint in both, the rear tooth force is the same. The difference between force on the anterior and posterior teeth is relatively greater in the long jawed fish.

Line 311 Likewise : “Here, we suggest that the mechanistic function of jaw elongation, may be to facilitate the back-fanged dentition with increased pressure/stress output RELATIVE TO THE ANTERIOR TEETH, facilitating (THE SEPARATION/DECOUPLING OF) prey manipulation and processeing.” See above (also note spelling error of processing)

Look at Figure 6 legend- lettering does not appear to match the diagram e.g. a is not edentulate etc.

Decision letter (RSOS-190040.R1)

02-Aug-2019

Dear Mr Mihalitsis:

On behalf of the Editors, I am pleased to inform you that your Manuscript RSOS-190040.R1 entitled "Functional implications of dentition-based morphotypes in piscivorous fishes" has been accepted for publication in Royal Society Open Science subject to minor revision in accordance with the referee suggestions. Please find the referees' comments at the end of this email.

The reviewers and Subject Editor have recommended publication, but also suggest some minor revisions to your manuscript. Therefore, I invite you to respond to the comments and revise your manuscript.

- Ethics statement

- Data accessibility

If you wish to submit your supporting data or code to Dryad (<http://datadryad.org/>), or modify your current submission to dryad, please use the following link:
<http://datadryad.org/submit?journalID=RSOS&manu=RSOS-190040.R1>

- Competing interests

- Authors' contributions

All submissions, other than those with a single author, must include an Authors' Contributions section which individually lists the specific contribution of each author. The list of Authors

should meet all of the following criteria; 1) substantial contributions to conception and design, or acquisition of data, or analysis and interpretation of data; 2) drafting the article or revising it critically for important intellectual content; and 3) final approval of the version to be published.

- Acknowledgements

- Funding statement

Because the schedule for publication is very tight, it is a condition of publication that you submit the revised version of your manuscript before 11-Aug-2019. Please note that the revision deadline will expire at 00.00am on this date. If you do not think you will be able to meet this date please let me know immediately.

- 1) A text file of the manuscript (tex, txt, rtf, docx or doc), references, tables (including captions) and figure captions. Do not upload a PDF as your "Main Document".
- 2) A separate electronic file of each figure (EPS or print-quality PDF preferred (either format should be produced directly from original creation package), or original software format)
- 3) Included a 100 word media summary of your paper when requested at submission. Please ensure you have entered correct contact details (email, institution and telephone) in your user account

4) Included the raw data to support the claims made in your paper. You can either include your data as electronic supplementary material or upload to a repository and include the relevant doi within your manuscript

5) All supplementary materials accompanying an accepted article will be treated as in their final form. Note that the Royal Society will neither edit nor typeset supplementary material and it will be hosted as provided. Please ensure that the supplementary material includes the paper details where possible (authors, article title, journal name).

on behalf of Professor Emily Standen (Associate Editor) and Kevin Padian (Subject Editor)
openscience@royalsociety.org

Associate Editor Comments to Author (Professor Emily Standen):

Dear Dr. Mihalitsis,

The most recent reviews for your manuscript entitled 'Functional implications of dentition-based morphotypes in piscivorous fishes' are very positive and there remain only a few small corrections and clarifications to be made. We will be happy to receive your paper with these comments addressed.

Thank you for your attention.

Emily

Reviewer comments to Author:
Reviewer: 2

Intro line 61- canines is misspelled
Line 81-82 establishing a functional link between a certain anatomical features and how (this) help(s) the organism perform a specific task - missing word

Methods- first line- remove comma after measured

Line 111- still have a problem with the term medially- teeth were angled lingually or labially- is that what you mean? I don't know what medially refers to in this case.

End of methods- "In this part of our study, we used a different set of morphological traits which were applicable to macrodont species exclusively. Traits used here were: variance in tooth sizes, smallest vs. largest tooth length of the five largest teeth, mean distance between five largest teeth, and largest tooth position. For a detailed description of each trait see Supplemental Table 1". Are these in addition to the forementioned morphological traits? If so, say so. Most readers won't go to the supplemental table.

Results line 194- in macrodont needs a space between words

In the discussion on lines 305-307

"If back fanged species represent a functional decoupling of the oral teeth, separating fast grabbing anterior teeth from slower but deeply penetrating posterior teeth, a longer lower jaw would maximize both the velocity advantage of the anterior tooth, and the (RELATIVE) force advantage of the posterior tooth."

The rear force advantage would be RELATIVE to the anterior velocity advantage- I don't think it would be an absolute. There would just be a relatively larger difference between front and back. If the adductor muscle is the same size developing the same force between two such fishes, and the rear tooth is the same distance from the jaw joint in both, the rear tooth force is the same. The difference between force on the anterior and posterior teeth is relatively greater in the long jawed fish.

Line 311 Likewise : "Here, we suggest that the mechanistic function of jaw elongation, may be to facilitate the back-fanged dentition with increased pressure/stress output RELATIVE TO THE ANTERIOR TEETH, facilitating (THE SEPARATION/DECOUPLING OF) prey manipulation and processing." See above (also note spelling error of processing)

Look at Figure 6 legend- lettering does not appear to match the diagram e.g. a is not edentulate etc.

Reviewer: 1

Comments to the Author(s)

The authors have tried to appease the reviewers from the last round, including myself. I think that they have done a good job. I do not know why they could not cite a few more fish papers on heterodonty in bony fishes, including those by younger authors (e.g., K. Bemis, K. Conway).....that would help the whole community as a whole, give them a hand when possible!

I look forward to seeing the final version.

Author's Response to Decision Letter for (RSOS-190040.R1)

See Appendix B.

Decision letter (RSOS-190040.R2)

13-Aug-2019

Dear Mr Mihalitsis,

I am pleased to inform you that your manuscript entitled "Functional implications of dentition-based morphotypes in piscivorous fishes" is now accepted for publication in Royal Society Open Science.

Kind regards,

on behalf of Professor Emily Standen (Associate Editor) and Kevin Padian (Subject Editor)
openscience@royalsociety.org

Follow Royal Society Publishing on Twitter: [@RSocPublishing](https://twitter.com/RSocPublishing)

Appendix A

Response to Editors and reviewers

Associate Editor's comments (Professor Emily Standen):

Associate Editor: 1

Comments to the Author:

Dear Dr. Mihalitsis,

We have now received both reviews of your manuscript entitled the Functional implications of dentition-based morphotypes in piscivorous fishes. In general the reviews are very positive and find the work interesting and valuable. Both reviewers do suggest clarifying the definition of homodonty in your paper as well as being more specific regarding the lack of literature on fish dentition. As one reviewer points out, there is a large body of literature on fish teeth and this should be recognized.

I look forward to seeing your careful address for the reviewers comments.

Sincerely,

Emily Standen

Associate Editor: 2

Comments to the Author:

Dear Michalis Mihalitsis,

Thank you for your submission of the manuscript entitled Functional implications of dentition-based morphotypes in piscivorous fishes. I find this an interesting topic and see value in broadening our ability to assess ecomorphology of basal vertebrates using teeth to compare with the tools that have been developed for mammalian teeth.

I am in agreement with the reviewer who suggests two main points that need to be addressed to increase the clarity and impact of your work. First, reconsider the use of the word homodonty, or at the very least expand your definition to help avoid confusion for readers, and second, incorporate more of the existing literature on fish teeth.

At this point we must reject your paper but we encourage you to fully address the reviewers comments and resubmit the paper for further review.

Emily Standen

Thank you for your constructive feedback. We have addressed all comments from both reviewers. Specifically, we have clearly defined homodonty in our manuscript, based on the works of Karel Liem:

"...In the majority of vertebrate species, the teeth, although they may differ in size, are structurally alike, a condition called homodont." (Liem et al. 2001)

Furthermore, we have expanded the text incorporating more work on fish dentition types from the literature.

We hope our revised manuscript meets your approval.

Best regards,

Michalis Mihalitsis (on behalf of the authors)

Editor comments:

Thanks for revising. As you can see, the first reviewer still has major concerns with the paper, especially with definitions and analyses. The second reviewer additionally points out that there is a considerable literature on this topic that is not acknowledged. This is a very important issue for us.

Please address all of the reviewers' comments in your revision. Best wishes.

Thank you for your constructive comments. We have now revised our manuscript and addressed all comments from reviewers. We believe these revisions have significantly increased the clarity of our manuscript, particularly with regards to reviewers concerns. We hope the manuscript meets your approval.

Best regards,

Michalis Mihalitsis (on behalf of the authors)

Comments to Author:

Reviewers' Comments to Author:

Reviewer: 1

Comments to the Author(s)

This is an interesting manuscript focused on function of "homodont" teleost fishes. Homodont of course is a term used in reference to a dentition comprised of a single tooth type (vs. heterodont).

The study is interesting and overall I am OK with much of the manuscript. However, I do have several issues with the introduction and also how the authors define homodont (which has implications for some of the authors arguments). I list my major issues below and then end the review with a few minor issues for the authors to take care of (minimal effort on their part).

Thank you for taking the time to carefully read our manuscript and your very constructive feedback. We have addressed each issue specifically below, including your concerns with regards to homodonty and the need to further elaborate on previous research in this field.

Major issues:

1. How do the authors define homodont? A fish with small conical teeth all of the same size (villiform for example) would be a textbook homodont in my opinion. But the authors focus on at least two species in which the teeth are of very different sizes (see there fig. 2c/d). This includes fishes with larger "canine" teeth either at the front or the rear of the jaws combined with smaller conical teeth. In order for these species to be homodont the canine teeth and the conical teeth would have to be the same type. Do the authors really consider these larger canines to be the same as the smaller conical teeth? One way to convince the reader of this would be to show that the canines were simply scaled up replicas of the smaller conical teeth (or visa versa). If they can not show this then I think this is a major flaw. These fishes would be heterodonts and not homodonts, which would require some restructuring of the introduction, and maybe discussion.

Thank you for this comment on this highly interesting matter. We agree with you that a villiform (as described above) would be a textbook homodont.

In our study we defined a homodont based on the works of Karel Liem (Liem et al. 2001), who defines homodonty as:

"...In the majority of vertebrate species, the teeth, although they may differ in size, are structurally alike, a condition called homodont."

and

"Although tooth size may vary, all the teeth of most fishes and most amphibians and reptiles have a similar shape, a condition termed homodont."

We have therefore more clearly explained our use of the term homodont in our manuscript. Please see lines 66-72 of our Introduction:

" Unlike mammals that have different shaped teeth (heterodont) (Ungar 2010), fishes and other lower vertebrates typically have similarly shaped teeth (homodont) (Hunter 1999). These descriptive terms, homodont and heterodont, need to be interpreted with caution, as the term 'different shaped teeth' can sometimes be misleading. In this study we follow (Liem et al. 2001), who said "...in the majority of vertebrate species, the teeth, although they may differ in size, are structurally alike, a condition called homodont." This issue was discussed by (D'Amore et al. 2019) who noted the need for a broad evaluation of tooth form and function."

2. The introduction is set up to convince the reader that there are almost no studies on the teeth of fishes, and those that are available are only descriptive. I think this is really not the

case and leads the reader a stray. For example, if one was to type "shark and teeth" or "teleost and teeth" into a search engine such as google scholar the number of returns would be >58,000 and >11,000, respectively. Of course, these numbers would include the descriptive studies that they authors make reference too but many, many "non-descriptive" studies on the teeth of fishes would also be included. In my opinion, the authors give these studies short shrift. I understand that the authors are trying to sale the study as a "first" (and in several aspects it is) but it is a little sneaky (maybe even deceitful) to try to convince the reader that there are almost no functional studies on the teeth of fishes. If one was to conduct a decent review, I am sure that there would be just as many studies on the teeth of fishes as there are on the teeth of mammals. My major recommendation is that the authors tone down the introduction.

We apologise for this dismissive appearance; this was not our intention. We have now incorporated further literature into our manuscript, especially in our Introduction. Please see lines 43-48 of our manuscript:

" Fishes, and more specifically, teleosts, constitute over half of all vertebrate species (Eschmeyer et al. 2010), however, our understanding of their oral tooth morphology was for a long time primarily at a descriptive level: small/large, conical/villiform/molariform (e.g.(Allen 1985)). In the last decade however, research has begun elucidating the morphology and potential function of several aspects of fish dentition (Grubich et al. 2008; Grubich et al. 2012; Ferguson et al. 2015; Corn et al. 2016; Galloway et al. 2016). These studies have provided invaluable information on how tooth morphology in fishes may influence their feeding capabilities."

Also, we changed some sentences to avoid confusion with regards to the amount of fish-based studies on teeth. Please see (lines 52-53):

"The lack of more precise/quantitative descriptions is not without good reason."

Changed to (lines 52-53):

"The limited number of more quantitative descriptions, when compared to mammalian dentition, is not without good reason."

Minor comments:

P3, L73: chondichthyans should be chondrichthyans

Thank you for pointing this out. Changed to chondrichthyans.

P4, L84: First mention of piscivorous. Rather abrupt....

Thank you for pointing this out. We have now added another paragraph above this section, explaining why piscivorous fishes were chosen as a study group and why they are important to study. Please see lines 86-91 of our manuscript:

" One group of organisms displaying high morphological diversity, and thus making them an ideal study group, are piscivorous coral reef fishes (Mihalitsis and Bellwood

2019). This group of fishes displays high morphological diversity related to feeding traits such as gape size (Goatley and Bellwood 2009; Mihalitsis and Bellwood 2019). It has been suggested that this diversity may reflect the potential for niche partitioning on different prey sizes, or different feeding modes (Mihalitsis and Bellwood 2019). However, before beginning to ask such questions, there is a need to first delineate the various dentition morphotypes found within this functional group."

P10, L "a villiform with a relatively large gape," does not make sense. The fish is not a villiform, it has a villiform dentition. Please revise.

Changed as suggested. This part of our manuscript now reads (lines 250-253):

"Mihalitsis and Bellwood [30] likewise found *Cephalopholis urodeta*, a piscivore with villiform dentition and a relatively large gape, captured prey head-first..."

P10, L242-244: "(see Supplemental Fig. 6 for dentition of *Varanus* monitor lizard, a species also appearing to be `back-fanged`, and other fish orders (e.g. Characiformes)." There is a problem with the brackets here. There are not enough or too many. Please correct. Also, seems a little out of context to provide a S figure of a lizard? The point could be made without the figure. Also, I can not even see the S figures.

This sentence has now been deleted. We wanted to show that the back-fanged dentition morphotype found here, can be found in other homodont vertebrate groups as well. We have now updated this figure with high quality images, and added it to the main text. Please see Fig. 7.

Supplementary material was not made available to the reviewer and would have been helpful.

We apologise that you were not able to see the Supplemental Material. We do not know why that was the case. Our Supplemental Material was submitted with the manuscript.

Reviewer: 2

Thank you for your insightful and constructive feedback. We have now revised our manuscript and incorporated all your comments and suggestions. Please see below.

Comments to the Author(s)

Good introduction. On line 81-82 you point out that homodont teeth have similar shapes but there might be other aspects of morphology that influence tooth function – at this point I would suggest you just elaborate on this a bit and add some verbiage to the effect "... including tooth size, position and occlusion patterns".

Thank you for this comment. We strongly agree with you that there are other aspects of tooth/dentition morphology which may be influencing tooth function (including the aspects you mention). However, this paragraph has now been deleted to reduce the focus of homodonty vs. heterodonty in our manuscript.

On a separate note, the problem with the homodont versus heterodont dichotomy is that sometimes there is a grey zone with what is termed homodont. It is clear that teleosts such as the drums have a heterodonty dentition with more molariform teeth posteriorly, but some other fishes have very recurved and pointy teeth anteriorly, and more triangular teeth throughout the jaw or posteriorly. Is this homodont? Heterodontus the shark is clearly heterodonty- but is the mako shark homodont or heterodont? Maybe you should very briefly mention in the intro that sometimes the terms are a bit vague and there are fishes that don't quite fit the dichotomy.

Thank you for this interesting comment. Indeed, we agree that such descriptive terms are not the best way to classify these dentition patterns. And since most of the teeth in such dentition types look alike, it becomes harder to make such classifications. Every tooth in the mouth of a fish will be morphologically different if examined at a sufficiently fine scale. However, to be able to infer and test the function of such dentition types, we need to start with broad morphological categories. To address this issue we have now followed your suggestion and added some text in our Introduction, further explaining this in more detail, emphasising that these terms are not always definitive. Please see lines 68-78 of our revised manuscript:

“However, these descriptive terms, homodont and heterodont, need to be interpreted with caution, as the term ‘different shaped teeth’ can sometimes be misleading. In this study we follow Liem (Liem et al. 2001), who noted that “...in the majority of vertebrate species, the teeth, although they may differ in size, are structurally alike, a condition called homodont”. This issue was discussed by (D’Amore et al. 2019) who noted the need for a broader evaluation of tooth form and function.

These terms (homodont/heterodont) offer definitions that provide a coarse framework for the comparative analysis of tooth form. However, tooth form may not be the only trait determining tooth function (Grubich et al. 2008; Ferguson et al. 2015) within lower vertebrates. Overlap among groups is inevitable. There is clearly a need to expand our frame of reference from individual tooth form and function to the entire dentition morphotype and its functional implications.”

In the methods line 96 do you mean vertical and horizontal oral gape distance? You just say “gapes”. It could be gape angle or distance. Clarify

We apologise for this lack of clarity. We meant gape distances. This part of our revised manuscript now reads (lines 105-106):

“ Vertical and horizontal oral gape distances were measured using scissors, following the methodology and definitions of Mihalitsis and Bellwood [30].”

Line 102 – what do you mean by “inwards”. Do you also mean the whole fish head was tilted at an angle to envision the teeth – and how was it tilted? You can’t just tilt the teeth. This is all unclear.

We apologise. Inwards here, refers to the angle of the teeth relative to the mouth of the fish. We have now edited this part of our manuscript for clarity and it now reads (lines 109-112):

“ While mouths were open, the left lower jaw was photographed laterally, the camera being perpendicular to the teeth. In species with villiform dentition, the teeth were found to be angled medially. Additional images were therefore taken with the camera at approximately at 45°, to capture the whole tooth length.”

Did you sometimes cut away a bit of labia (gums) to expose the total teeth? Sometimes the “gum” hides the base of the teeth and you can’t see the total length nor the base. We do this sometimes.

Yes, we did. We apologise for not mentioning this. If the species examined had an enlarged lip, the lip was pulled down and fixed with a pin to reveal the whole tooth length. We have now added this explanation to our manuscript, and it now reads (lines 112-114):

“ In species with enlarged lips, the lips were pulled downwards and fixed with a pin to reveal the full length of the teeth.”

Line 105 – I do disagree that you term fish with very small teeth as edentulate. To me you need another name for this. Cypriniforms are edentulate. Some fish have very small villiform teeth; you could even call them cardiform, but NOT edentulate.

Thank you for this comment. We had not heard of the term cardiform before. According to the Meriam-Webster Dictionary cardiform means “arranged like a series of combs or wool cards (as the teeth of certain fishes).”, and in Fishbase as “ Small, sharp, slender teeth, like those on wool cards; arranged like a series of combs or wool cards.”.

This definition appears to be very similar to villiform dentition types based on the definition provided on Fishbase:

“ Teeth so slender and crowded as to resemble bristles of a brush, so that it is difficult or impossible to number them in terms of rows; small slender teeth that form velvety bands.”

We are therefore unsure what the difference between these two dentitions would be.

With the term edentulate, we describe dentition types where teeth were so small that even a DSLR camera with a macro lens was unable to capture their length. We now explain that in this study, 'edentulate' refers to species with no teeth, or teeth that are too small to detect. Please see lines 114-117:

" Some species (e.g. Neoniphon sammara) have numerous teeth, however, they are so small (generally <1mm) and compact to be almost invisible to the naked eye; for the purposes of this study they were classified as edentulate as they were too small to measure."

In the Analysis section here are some comments: You need to change the terminology largest to longest (throughout) if that is truly what you mean. Smallest should also be changed to shortest if that is what you mean. Large and small refers to size, which is different from long and short.

Thank you for this comment. We understand your concern with this terminology. However, as the species analysed in our study are generally considered homodont, they have similarly shaped teeth at the scale investigated. Therefore, the largest tooth, is by definition, the longest as well, and the smallest is the shortest. To avoid confusion and the general assumption that long may mean thin, we retained the largest and smallest (while explaining that this does mean longest and shortest). We have now added a statement in our Materials and Methods section clarifying this (lines 128-129):

" Throughout our manuscript, the terms largest and smallest teeth is based on tooth length, and therefore also refer to longest and shortest teeth respectively (given the similarity in tooth shapes)."

Line 113 largest tooth width "at the base"

Changed as suggested.

Line 111 Measured units were : lower jaw length...

Changed as suggested

What about upper jaw teeth? If they mirror lower jaw teeth say so. Some fishes have extremely large upper jaw teeth and your analysis does not account for that (gempylids).

Thank you for this useful comment. Unfortunately, upper jaw teeth were not quantified in this study, as we wanted to focus on the dentition directly associated to jaw closing biomechanics. However, based on qualitative observations, we noticed that villiform and macrodont morphotypes, in almost all cases, appeared to have a larger tooth than the rest (upper jaw teeth) anteriormost to the upper jaw, followed by smaller teeth of

similar size. We never observed a tooth larger than the rest (in the upper jaw) being at the posterior part of the upper jaw. We have now added this observation to our manuscript (Materials & Methods, Results) as well as discussing the functional implications.

Please see sections below of our manuscript:

Materials and Methods (line 114):

“Qualitative observations on the upper jaw dentition patterns were also made.”

Results (lines 187-190):

“ Upper jaw dentition in villiform and macrodont morphotypes were primarily described by a large caniniform tooth on the anteriormost margin of the premaxilla (usually smaller in species with villiform dentition), followed posteriorly by smaller similarly-shaped teeth (e.g. Fig. 6).”

Discussion (lines 283-288):

“ We suggest that back-fanged morphotypes, could be exhibiting a form of functional decoupling, with the anteriormost canines (higher speed/less force) being used for grabbing prey whereas posteriorly positioned canines (lower speed/higher force) are used for post capture processing (Fig. 4), such as deeply piercing and/or lacerating prey, especially when using the slashing behaviour described above. This is further highlighted by the lack of back-fanged tooth distributions on the upper jaw (Fig. 6).”

Lastly, we have added a Figure in our manuscript to further illustrate this. Please see Figure 6 with the following caption:

“ Fig. 6. Full dentition morphotypes (both upper and lower jaws) displaying inferred functional capabilities based on biomechanical properties. a) edentulate, b) villiform, c) front-fanged macrodont, d) back-fanged macrodont. Note the absence of the back-fanged dentition in upper jaws.”

Line 138 – “vertical gape height” you say gape size which is not specific and on that line you say “mouth shape” what does that mean, be specific. Do you mean circular, notched, laterally occluded?

We apologise for this lack of clarity. Gape sizes were measured following (Mihalitsis and Bellwood 2017) who defined specific morphological gapes, and then tested which of these gapes best described maximum prey size.

We have now clarified gape sizes to mean vertical oral gape and horizontal oral gape (see lines 152-153).

Mouth shape was defined as the ratio between vertical oral gape to horizontal oral gape and has now been added to the manuscript (see line 153).

Line 150 and other similar sections- you say largest tooth position- I think you mean longest tooth position. If you mean largest then you must define what constitutes "largest" – is that base width and length, height also, what? Do a global search for largest, large, small, smallest and replace with the correct terms.

Thank you for this comment. However, as noted above the species analysed in our study are primarily considered homodont, meaning they have similarly shaped teeth at the scales investigated. Therefore, the largest tooth is, by definition the longest as well, and the smallest is the shortest. This has been clarified in the text.

Please see lines 128-129 of our manuscript: " Throughout our manuscript, the terms largest and smallest teeth is based on tooth length, and therefore also refer to longest and shortest teeth respectively (given the similarity in tooth shapes)."

On figure 1 I now see horizontal oral gape and vertical oral gape. This needs to be spelled out in the analysis sections. Was gape size measured at the anterior tips of the upper and lower jaws? Some researchers have measured it at the notch part of the gape.

Thank you for pointing this out. We recently wrote a paper focusing solely on gapes. You are right in that multiple studies have used different morphological measurements to measure 'gape'. For more detail on this please see Introduction and Table 1 of (Mihalitsis and Bellwood 2017).

Specific morphological measurements for gape sizes were originally mentioned in the Supplemental Material, however have now been also added to the main text. Please see lines 152-153 of our revised manuscript.

Results

Line 168- remove qualifiers like "highly" – just say similar. The problem with this is, how "high" is highly?

Changed as suggested. This part of the manuscript now reads (lines 184-186):

" Our Principal Coordinate Analysis (PCoA) revealed similar results to our PPCA, suggesting zero values of edentulates had minimal effect on our analysis (see Supplemental Fig. 2)."

Line 179 – oval mouths- do you mean notched?

This description relates to mouth shape, measured by vertical oral gape/horizontal oral gape (ratio). Here, a value of 1 means that vertical oral gape = horizontal oral gape, and therefore the mouth is round in shape when fully open (viewed anteriorly). A value >1 means that the vertical gape is greater than the horizontal gape and therefore the mouth shape is oval-shaped along the vertical axis. A value <1 means that vertical

gape is smaller than the horizontal gape and therefore the mouth shape is oval-shaped along the horizontal axis.

What about protrusion characteristics? Does the macrodont tooth type have generally less protrusion than the villiform/edentulate forms? That wasn't clear in the text.

We have now added a sentence in our results to address this. Please see lines (199-201).

“ Jaw protrusion appears to be mostly associated with edentulate and villiform morphotypes; this was, however, not significant in PGLS models (Supplemental Table 4).”

Line 194- located anteriorly ON the jaw- it cant be anterior to the jaw, nor posterior to the jaw, so correct those sentences.

Changed as suggested.

Line 201 –unevenly sized

Changed as suggested.

I really liked how you summarized each type with a clear description following the details- nice job.

Thank you for this positive feedback!

Discussion

Lines 234-236- are you referring to suction versus ram capture? Engulfing and grabbing are not terms we usually use in the prey capture literature. What is also important here is the difference between getting the prey to the mouth, and then processing or holding the prey in the jaws. You need to be clearer regarding this difference. Edentulous species or those with villiform/cardiform teeth may use some degree of suction to capture the prey and the fine teeth to hold it (hence the rounder mouth aperture); macrodents tend to use more ram (barracuda), grasping and cutting the prey with the enlarged teeth. However, there is more than one way to skin a cat (many to one mapping-Wainwright) and wahoo have relatively small teeth, all about the same size, and use incredible ram. Then some fish such as grouper will use more ram or suction (with relatively small teeth) depending on where the prey are (Collins & Motta 2017 A kinematic investigation into the feeding behavior of the Goliath Grouper *Epinephelus itajara*. *Environ Biol Fish* 100: 309-323.)

We have now changed this part of our manuscript by expanding the terms engulf and grab, to consider ram and suction. Furthermore, we have also cited the paper suggested. Please see lines 253-258 of our manuscript:

“ Based on the observed morphologies and behaviour, we suggest that edentulate and villiform species, with larger gape sizes, might be more efficient in ‘engulfing’ their prey through ambush predation, whereas macrodonts, with smaller gape sizes but larger teeth, might be more efficient at ‘grabbing’ their prey after a short-distance lunge and/or longer pursuit. This axis of variation may also reflect varying contributions from suction vs. ram in engulfing vs. grabbing species (Ferry et al. 2015; Longo et al. 2016; Collins and Motta 2017).”

The section ending on line 25 brings to mind the study on tooth pressure by macrodont piscivores- Ferguson et al 2015 Feeding performance of king mackerel J Exp Zool 323A:399-413. These extremely sharp teeth, especially at the rear of the jaw, exert extreme pressure (force/unit area) on the prey. You might discuss that. This could also fit after the section ending on line 265. I believe there is one other fish study that correctly calculated bite pressure – it may be Westneat or Anderson. Most confuse force and pressure- and all the pressure stuff on line is crap!

Thank you for pointing out this paper. This is really interesting and highly relevant to our study, as well as our future work. However, we are unsure which section you are referring to, as line 25 is part of the Abstract (perhaps line 250?). Nevertheless, we have now referenced this paper in two sections of our manuscript. Please see lines

(267-269): Teeth used to penetrate prey, are highly linked to the biomechanical property of stress (e.g.(Whitenack et al. 2011)) , that is, the force applied to an object relative to the area over which it is applied ($\sigma = \text{force/area}$, SI= newton/meter²) (Vogel 2013; Ferguson et al. 2015).

(278-282):“ By having a large caniniform tooth posteriorly in their jaw, these species gain, on average, a 42 % force advantage when compared to a same sized tooth positioned at the anterior-most point of the jaw (Fig. 4) (see also Ferguson et al. 2015). This value mirrors differences reported in anterior vs. posterior jaw bite pressure, calculated in (Ferguson et al. 2015) for King Mackerel (Scomberomorus cavalla).”

Really good point on lines 282-286 – reminds me of the diminutive Pike Killifish *Belonesox belizanus* - a really weird killifish- look at the pictures.

Thank you, again, for pointing out this fish species. By looking further into this species, we found some interesting research which has also investigated the relationship between lower jaw length and teeth. We have now incorporated these studies and changed this part of our manuscript. Please see lines 305-313:

“ If back fanged species represent a functional decoupling of the oral teeth, separating fast grabbing anterior teeth from slower but deeply penetrating posterior teeth, a longer lower jaw would maximize both the velocity advantage of the anterior tooth, and the force advantage of the posterior tooth. Interestingly, lower jaw elongation has

arisen on multiple occasions and has been widely associated with increased piscivory (Fryer and Iles 1972; Barnett et al. 2006; McGee et al. 2015). It has been suggested that the mechanistic function underlying jaw elongation is an increase in gape size, and creating a larger contact area between predator and prey for prey manipulation (Grubich et al. 2008; Ferry-Graham et al. 2010). Here, we suggest that the mechanistic function of jaw elongation, may be to facilitate the back-fanged dentition with increased pressure/stress output, facilitating both prey manipulation and processeing.”

Rather than the ending paragraph I would suggest a summary paragraph – a lot of the time the reader zeros in on the final summary of the findings and main conclusions. Yours is a bit too general and the punch line is lost.

Thank you for this suggestion. We have now added a summary paragraph. Please see lines 316-324 of our manuscript:

“ Overall, we provide a quantitative framework for identifying dentition morphotypes in lower vertebrates, especially piscivorous fishes, and provide a putative functional interpretation of these distinct morphotypes. We identify three distinct dentition morphotypes (edentulate/villiform/macrodont) that appear to be encompassed by just two functional groups, broadly classified as ‘engulfers’ vs. ‘grabbers’. Also, within macrodents we identify a continuum between front-fanged and back-fanged species, and explore the functional implications separating teeth involved in procurement (grabbing) vs. processing (laceration). We highlight the potential for functional decoupling in fish teeth, based not on the shape of the tooth, but their relative position along the jaw.”

References

- Allen GR (1985) FAO species catalogue vol. 6 snappers of the world: An annotated and illustrated catalogue of Lutjanid species known to date. Food and Agriculture Organization of the United Nations
- Barnett A, Bellwood DR, Hoey AS (2006) Trophic ecomorphology of cardinalfish. *Marine Ecology Progress Series* 322:249-257
- Collins A, Motta P (2017) A kinematic investigation into the feeding behavior of the Goliath grouper *Epinephelus itajara*. *Environmental biology of fishes* 100:309-323
- Corn KA, Farina SC, Brash J, Summers AP (2016) Modelling tooth–prey interactions in sharks: the importance of dynamic testing. *Royal Society open science* 3:160141
- D’Amore DC, Harmon M, Drumheller SK, Testin JJ (2019) Quantitative heterodonty in Crocodylia: assessing size and shape across modern and extinct taxa. *PeerJ* 7:e6485
- Eschmeyer WN, Fricke R, Fong JD, Polack DA (2010) Marine fish diversity: history of knowledge and discovery (Pisces). *Zootaxa* 2525:19-50
- Ferguson AR, Huber DR, Lajeunesse MJ, Motta PJ (2015) Feeding performance of king Mackerel, *Scomberomorus cavalla*. *Journal of Experimental Zoology Part A: Ecological Genetics and Physiology* 323:399-413
- Ferry-Graham LA, Hernandez LP, Gibb AC, Pace C (2010) Unusual kinematics and jaw morphology associated with piscivory in the poeciliid, *Belonesox belizanus*. *Zoology* 113:140-147
- Ferry LA, Paig-Tran EM, Gibb AC (2015) Suction, ram, and biting: deviations and limitations to the capture of aquatic prey. *Integrative and comparative biology* 55:97-109
- Fryer G, Iles T (1972) Cichlid fishes of the great lakes of Africa

- Galloway KA, Anderson PS, Wilga CD, Summers AP (2016) Performance of teeth of lingcod, *Ophiodon elongatus*, over ontogeny. *Journal of Experimental Zoology Part A: Ecological Genetics and Physiology* 325:99-105
- Goatley C, Bellwood DR (2009) Morphological structure in a reef fish assemblage. *Coral Reefs* 28:449-457
- Grubich JR, Rice AN, Westneat MW (2008) Functional morphology of bite mechanics in the great barracuda (*Sphyraena barracuda*). *Zoology* 111:16-29
- Grubich JR, Huskey S, Crofts S, Orti G, Porto J (2012) Mega-Bites: Extreme jaw forces of living and extinct piranhas (Serrasalminae). *Scientific reports* 2:1009
- Hunter JP (1999) Teeth: evolution of complex teeth. In: Singer R (ed) *Encyclopedia of Paleontology*. Dearborn Publishers, Fitzroy,
- Liem KF, Bemis WE, Walker WF, Grande L (2001) Functional anatomy of the vertebrates: an evolutionary perspective
- Longo SJ, McGee MD, Oufiero CE, Waltzek TB, Wainwright PC (2016) Body ram, not suction, is the primary axis of suction-feeding diversity in spiny-rayed fishes. *Journal of Experimental Biology* 219:119-128
- McGee MD, Borstein SR, Neches RY, Buescher HH, Seehausen O, Wainwright PC (2015) A pharyngeal jaw evolutionary innovation facilitated extinction in Lake Victoria cichlids. *Science* 350:1077-1079
- Mihalitsis M, Bellwood DR (2017) A morphological and functional basis for maximum prey size in piscivorous fishes. *PloS one* 12:e0184679
- Mihalitsis M, Bellwood DR (2019) Morphological and functional diversity of piscivorous fishes on coral reefs. *Coral Reefs*
- Ungar PS (2010) *Mammal teeth: origin, evolution, and diversity*. JHU Press
- Vogel S (2013) *Comparative biomechanics: life's physical world*. Princeton University Press
- Whitenack LB, Simkins Jr DC, Motta PJ (2011) Biology meets engineering: the structural mechanics of fossil and extant shark teeth. *Journal of morphology* 272:169-179

Appendix B

Response to Reviewers

Dear Mr Mihalitsis:

On behalf of the Editors, I am pleased to inform you that your Manuscript RSOS-190040.R1 entitled "Functional implications of dentition-based morphotypes in piscivorous fishes" has been accepted for publication in Royal Society Open Science subject to minor revision in accordance with the referee suggestions. Please find the referees' comments at the end of this email.

The reviewers and Subject Editor have recommended publication, but also suggest some minor revisions to your manuscript. Therefore, I invite you to respond to the comments and revise your manuscript.

Kind regards,

Alice Power

Editorial Coordinator

on behalf of Professor Emily Standen (Associate Editor) and Kevin Padian (Subject Editor)

Associate Editor Comments to Author (Professor Emily Standen):

Dear Dr. Mihalitsis,

The most recent reviews for your manuscript entitled 'Functional implications of dentition-based morphotypes in piscivorous fishes' are very positive and there remain only a few small corrections

and clarifications to be made. We will be happy to receive your paper with these comments addressed.

Thank you for your attention.

Emily

Dear Prof. Standen,

Thank you for your feedback and constructive comments. All comments have now been addressed below.

Best regards (on behalf of the authors)

Michalis Mihalitsis

Reviewer comments to Author:

Reviewer: 2

Thank you for reading our manuscript. All comments have been addressed below.

Intro line 61- canines is misspelled

Thank you. Changed to canines.

Line 81-82 establishing a functional link between a certain anatomical features and how (this) help(s) the organism perform a specific task – missing word

Thank you. Changed to: " i.e. establishing a functional link between certain anatomical features and how they help the organism perform a specific task (e.g. feeding)."

Methods- first line- remove comma after measured

Changed as suggested.

Line 111- still have a problem with the term medially- teeth were angled lingually or labially- is that what you mean? I don't know what medially refers to in this case.

Changed to " In species with villiform dentition, the teeth were found to be angled medially (lingually)."

End of methods- "In this part of our study, we used a different set of morphological traits which were applicable to macrodont species exclusively. Traits used here were: variance in tooth sizes, smallest vs. largest tooth length of the five largest teeth, mean distance between five largest teeth, and largest tooth position. For a detailed description of each trait see Supplemental Table 1". Are these in addition to the forementioned morphological traits? If so, say so. Most readers won't go to the supplemental table.

We apologise for this not being clear. Some morphological measurements were used exclusively in our first morphological analysis (Fig. 1) (e.g. number of tooth rows), others exclusively in our second morphological analysis (e.g. mean distance between teeth) (Fig. 3), and some were used in

both analyses (e.g. tooth length). For ease of interpretation we have added the vector arrows showing traits used for each analysis in our Figures (see Fig. 1 and Fig. 3). To clarify, we have now re-written this part of our manuscript, and it now reads (lines 164-173):

“ During initial analysis, we found that some morphological traits did not conform with morphotype divisions. For example, largest tooth position (relative to jaw length) is uninformative for villiform and edentulate fish, as villiform fish have highly homogenous tooth sizes along their jaw (e.g. Fig. 2), and edentulate fish teeth are either exceedingly small teeth or absent. We therefore undertook a second morphological trait-based analysis where we included only macrodont species (i.e. excluding villiform and edentulate species). In this part of our study, we used a different set of morphological traits which were applicable to macrodont species exclusively. The traits used in the analysis of macrodont species were: variance in tooth sizes, smallest vs. largest tooth length of the five largest teeth, mean distance between five largest teeth, and largest tooth position. For a detailed description of each trait see Supplemental Table 1.”

Results line 194- in macrodont needs a space between words

Changed as suggested.

In the discussion on lines 305-307

“If back fanged species represent a functional decoupling of the oral teeth, separating fast grabbing anterior teeth from slower but deeply penetrating posterior teeth, a longer lower jaw would maximize both the velocity advantage of the anterior tooth, and the (RELATIVE) force advantage of the posterior tooth.”

The rear force advantage would be RELATIVE to the anterior velocity advantage- I don't think it would be an absolute. There would just be a relatively larger difference between front and back. If the adductor muscle is the same size developing the same force between two such fishes, and the rear tooth is the same distance from the jaw joint in both, the rear tooth force is the same. The difference between force on the anterior and posterior teeth is relatively greater in the long jawed fish.

Changed to: *“ a longer lower jaw would maximize both the velocity advantage of the anterior tooth, and the force advantage of the posterior tooth (relative to the anterior tooth).”*

Line 311 Likewise : “Here, we suggest that the mechanistic function of jaw elongation, may be to facilitate the back-fanged dentition with increased pressure/stress output RELATIVE TO THE ANTERIOR TEETH, facilitating (THE SEPARATION/DECOUPLING OF) prey manipulation and processeing.” See above (also note spelling error of processing)

Changed to: *“ Here, we suggest that the mechanistic function of jaw elongation, may be to facilitate the back-fanged dentition, with increased pressure/stress output relative to the anterior teeth, thus facilitating prey manipulation and processing.”*

Look at Figure 6 legend- lettering does not appear to match the diagram e.g. a is not edentulate etc.

Thank you for pointing this out. We have now changed the lettering to match the diagram.

Reviewer: 1

Comments to the Author(s)

The authors have tried to appease the reviewers from the last round, including myself. I think that they have done a good job. I do not know why they could not cite a few more fish papers on heterodonty in bony fishes, including those by younger authors (e.g., K. Bemis, K. Conway).....that would help the whole community as a whole, give them a hand when possible!

I look forward to seeing the final version.

Thank you for your comments and feedback. We have now included research from the suggested authors. Please see lines 45-46 of our manuscript:

“ In the last decade, however, research has begun to elucidate the morphology and potential function of several aspects of fish dentition (Grubich et al. 2008; Grubich et al. 2012; Bellwood et al. 2014; Conway et al. 2015; Ferguson et al. 2015; Corn et al. 2016; Galloway et al. 2016; Bemis et al. 2019).”

References

- Bellwood DR, Hoey AS, Bellwood O, Goatley CH (2014) Evolution of long-toothed fishes and the changing nature of fish–benthos interactions on coral reefs. *Nature Communications* 5:3144
- Bemis KE, Burke SM, St. John CA, Hilton EJ, Bemis WE (2019) Tooth development and replacement in the Atlantic Cutlassfish, *Trichiurus lepturus*, with comparisons to other Scombroidei. *Journal of morphology* 280:78-94
- Conway KW, Bertrand NG, Browning Z, Lancon TW, Clubb Jr FJ (2015) Heterodonty in the New World: an SEM investigation of oral jaw dentition in the clingfishes of the subfamily Gobiesocinae (Teleostei: Gobiesocidae). *Copeia* 103:973-998
- Corn KA, Farina SC, Brash J, Summers AP (2016) Modelling tooth–prey interactions in sharks: the importance of dynamic testing. *Royal Society open science* 3:160141
- Ferguson AR, Huber DR, Lajeunesse MJ, Motta PJ (2015) Feeding performance of king Mackerel, *Scomberomorus cavalla*. *Journal of Experimental Zoology Part A: Ecological Genetics and Physiology* 323:399-413
- Galloway KA, Anderson PS, Wilga CD, Summers AP (2016) Performance of teeth of lingcod, *Ophiodon elongatus*, over ontogeny. *Journal of Experimental Zoology Part A: Ecological Genetics and Physiology* 325:99-105
- Grubich JR, Rice AN, Westneat MW (2008) Functional morphology of bite mechanics in the great barracuda (*Sphyraena barracuda*). *Zoology* 111:16-29
- Grubich JR, Huskey S, Crofts S, Orti G, Porto J (2012) Mega-Bites: Extreme jaw forces of living and extinct piranhas (Serrasalminae). *Scientific reports* 2:1009